# METAGEN: A DSL, DATABASE, AND BENCHMARK FOR VLM-ASSISTED METAMATERIAL GENERATION

## ABSTRACT

Metamaterials are micro-architected structures whose geometry imparts highly tunable—often counter-intuitive—bulk properties. Due to their geometric complexity and a non-trivial mapping from architecture to behaviour, metamaterial design is difficult and inaccessible for non-experts. Vision-language models (VLMs) offer a promising platform for general, democratized metamaterial design, but no existing metamaterial representations or datasets facilitate the creation or evaluation of such an Assistant. We address these challenges with three complementary contributions. **(i)** *MetaDSL*: a compact, semantically rich domain-specific language (DSL) that captures diverse metamaterial designs in a form that is tailored for both humans and VLMs. **(ii)** *MetaDB*: a curated repository of more than $150\,000$ parameterized MetaDSL programs together with their derivatives—three-dimensional geometry, multi-view renderings, and simulated elastic properties. **(iii)** *MetaBench*: benchmark suites that probe three core capabilities of VLM design assistants: structure reconstruction, property-driven inverse design, and performance prediction. We use MetaBench to evaluate the performance of three state-of-the-art VLMs and two fine-tuned variants. When paired with case studies, our evaluations show that this ecosystem provides a strong foundation for VLM-assisted metamaterial design and learned structure–representation–property relationships.

## 1 INTRODUCTION

*Metamaterials* represent a key frontier in materials science: by exploiting small, patterned geometries, they endow bulk materials with properties beyond those of the constituent substance. Careful geometric tuning yields extraordinary behaviours such as programmable deformation (Jenett et al., 2020; Babaee et al., 2013), extreme strength-to-weight ratios (Qin et al., 2017), and seemingly contradictory properties, like stiffness and stretchiness (Surjadi et al., 2025). With such sprawling possibilities, one might expect to see metamaterials in all corners of engineering design, improving biomedical implants (Ataee et al., 2018; Ambu & Morabito, 2019), civil infrastructure (Alavi, 2025), energy management (Fan et al., 2022; Attarzadeh et al., 2022), and more. However, existing design tools require deep domain expertise, rendering them inaccessible to most potential users (Cui et al., 2024). Thus, to unlock the potential of metamaterials, it is necessary to democratize their design.

AI-powered design assistants are key to democratized metamaterial design, as they allow non-expert users to navigate the complex structure–property relationship that lies at the core of metamaterial design. In its most automated form, an assistant would identify a suitable structure based on a set of user-provided performance specifications (*inverse design*) (Cui et al., 2024; Mirramezani et al., 2025; Yang et al., 2024a; Maurizi et al., 2025). To facilitate user-guided exploration, the assistant should also be able to realize a functional representation of a user-described structure (*reconstruction*) (Tian et al., 2025; Naghavi Khanghah et al., 2024) and predict its behavior (*material understanding*) (Chen et al., 2025; Meyer et al., 2022). Although each task is independently well-studied and facilitated by data-driven AI techniques (Lee et al., 2024), no previous works have targeted a high-level, general-purpose design assistant capable of all three tasks.

Vision-language models (VLMs) offer an ideal platform for AI-assisted metamaterial design, as they excel at the cross-modal reasoning and generation required by the tasks above, while providing a natural, conversational interface suitable for non-experts. However, *realizing* our design assistant poses several challenges. First, it requires a metamaterial representation suitable for conveying

complex, heterogeneous geometry between humans and VLMs. Second, it requires a large, diverse, high-quality dataset featuring intuitive, verifiable formulations for each of our three target tasks. Existing options for both requirements proved insufficient, due to severe fragmentation within existing metamaterial literature (Section 2).

To address these issues and lay the foundation for VLM-assisted metamaterial design, we introduce a general, extensible ecosystem (MetaGen), anchored by 3 components:

1. **MetaDSL**: a domain-specific language (DSL) that captures 3D metamaterials in a structured, compact, and expressive form, which was jointly tailored for the needs of human designers, VLMs, and algorithmic generation/manipulation.

2. **MetaDB**: a database of more than $150\,000$ metamaterials, each pairing a MetaDSL program with the derived 3-D geometry, rendered images, and simulated properties.

3. **MetaBench**: a benchmark targeting three task categories – reconstruction, inverse design, and material understanding – using data sampled from MetaDB.

We use MetaBench to evaluate 3 off-the-shelf VLMs and 2 finetuned variants. Even with simple training techniques, our finetuned models demonstrate promising performance on metamaterial design tasks – including the first accessible, multi-architecture inverse-design approach that is mesh- and simulation-free at inference time. This represents a fundamental contribution to metamaterials. However, it also expands the scope of VLM-assisted reverse engineering, by demonstrating VLMs' ability to traverse not one but *two* complex domain mappings (code$\leftrightarrow$geometry$\leftrightarrow$performance) across three separate modalities.

Each component of MetaGen is designed for extensibility and community contribution, such that they can evolve seamlessly alongside the state of the art in materials science and agentic design. Collectively, our ecosystem provides a coherent, extensible knowledge base for metamaterial design, while laying the foundation for intuitive, efficient human–AI collaboration in architected materials.

## 2 BACKGROUND

**Metamaterial Representation** A metamaterial's behavior is primarily governed by its *architecture* – the configuration of solid material(s) and voids within a designated volume (Schaedler & Carter, 2016). This design space is infinitely large and heterogeneous, so researchers rely on shape parameterizations to expose useful architectures (Lee et al., 2024). Some works hand-design structure templates with a few exposed parameters (Muhammad & Lim, 2021; Frenzel et al., 2017; Meier et al., 2025). Parametric (multi-)classes use class-specific primitives and rules to describe entire families of related topologies, yielding thousands of trusses (Panetta et al., 2015; Abu-Mualla & Huang, 2024), plate lattices (Sun et al., 2023a) and curved shells (Xu et al., 2023; Liu et al., 2022). Implicit functions, signed distance fields (SDFs), and constructive solid geometry (CSG) are also common for structures based on triply periodic minimal surfaces (TPMS) (Yang & Buehler, 2022; Chan et al., 2020; Fisher et al., 2023), simple volumetric primitives (Babaee et al., 2013), and physical phenomena (Kumar et al., 2020). This plurality persists because each representation is uniquely well-suited to a particular subspace. Nevertheless, these siloed, class-specific parametrizations fragment the field (Lee et al., 2024), limit exploration (Berger et al., 2017), impede dataset reuse and extensibility (Lee et al., 2024), and preclude emerging multi-class hybrids (Surjadi et al., 2025; Chen et al., 2019; White et al., 2021). MetaDSL preserves the power of each approach, by allowing them to coexist in a unified interface.

Non-parametric encodings (e.g., voxel grids) also address siloing by capturing diverse architectures, yielding some of the largest, most diverse datasets, with $140k - 180k$ structures (Yang et al., 2024a; Xue et al., 2025). However, voxels are inferior for many applications due to their limited precision, which misrepresents structure features (Xue et al., 2025) and degrades mechanical performance through stress concentrations that persist at all resolutions (Thandaga Nagaraju et al., 2022). A hybrid voxel-SDF mitigates these issues (Xue et al., 2025). However, neither approach offers high-level navigability: voxel space is combinatorial with many invalid (disconnected) designs (Zhu et al., 2017), most grids are seeded by compact class-representations (Yang et al., 2024a; Xue et al., 2025), and missing semantic data complicates human-in-the-loop design (Makatura et al., 2023) and analysis (Chen et al., 2018). MetaDSL prioritizes semantic information, compactness, and geometric precision, without sacrificing expressiveness.

Recent work by Makatura et al. (2023) also balances expressiveness, precision, and interpretability through a procedural metamaterial graph (ProcMeta). Their workflow uses a graphical user interface (GUI) to build a representative piece of the structure within a "fundamental bounding volume" (FBV); then, the FBV and its contents undergo spatial transformations to complete the structure. The ProcMeta graph preserves the representative structure and symmetry operations. However, it omits the concept of an FBV in favor of unconstrained coordinates. This obscures important semantic information, and reduces the graph's utility outside of interactive GUI. Many other generators leverage FBV-like ideas for symmetric structures (Panetta et al., 2015; Abu-Mualla & Huang, 2024; Mirramezani et al., 2025; Lumpe & Stankovic, 2021), but each output representation discards or externalizes at least some part of the process information. MetaDSL elevates FBVs to a core language construct, encodes design constraints through checkable types, and preserves a complete description of the construction process. MetaDSL also exceeds the expressive power of prior FBV-based representations by accommodating non-skeleton specifications like implicit functions.

**Metamaterial Design**   Metamaterial design is performance-driven: forward design builds metamaterial structures based on expert intuition (Surjadi et al., 2025; Babaee et al., 2013; Jenett et al., 2020), while inverse design automatically generates suitable structures (Ha et al., 2023; Panetta et al., 2017). Topology optimization (TO) is a common inverse technique, but it requires one differentiable objective, many expensive simulations, and limited-resolution domains (Schumacher et al., 2015; Zhu et al., 2017). Data-driven methods learn pertinent relationships directly (Lee et al., 2024) by leveraging techniques such as diffusion over pixels/voxels (Yang et al., 2024a; Bastek & Kochmann, 2023), random forests (Chen et al., 2025), and graph neural networks (Maurizi et al., 2025; Meyer et al., 2022). VLMs have been used for specific subtasks, like producing natural language descriptions of 2D metamaterials (Tian et al., 2025) and inferring edge connections over partial material graphs (Naghavi Khanghah et al., 2024). However, nearly all of these approaches are restricted to 2D or a particular class – restricting the attainable property gamut (Berger et al., 2017) – and one of the three tasks (reconstruction, material understanding, or inverse design). We are the first to target all three tasks for 3D metamaterials, through a general VLM-assistant based on intuitive, high-level inputs. For 3D reconstruction and property prediction, we operate on images. Inverse design begins with a multi-property profile expressed in natural language ("auxetic metamaterial that is stiff along the $y$ direction"); this permits general multi-objective queries, with a lower input burden than e.g. elasticity tensors (Kumar et al., 2020) or stress-strain curves (Maurizi et al., 2025).

**Vision–Language Models (VLMs) for Design**   VLMs have permeated computational design, including 3-D scene design (Yang et al., 2024b; Kumaran et al., 2023), mesh generation and editing (Sun et al., 2023b; Wang et al., 2024; Jones et al., 2025; Huang et al., 2024; Yamada et al., 2024), interior layouts (Çelen et al., 2024), sewing-pattern synthesis (Nakayama et al., 2025; Bian et al., 2025), and computer-aided engineering (Makatura et al., 2024a;b; Choi et al., 2025; Yuan et al., 2024; Picard et al., 2025). VLMs are ideal for the long-standing, ill-posed challenges in these domains, such as property prediction (Adak et al., 2025) and the inference of procedural models from images and text (Li et al., 2025; Khan et al., 2024). Most works use domain-specific procedures (code) as the medium, with novel grammars and specialist knowledge accommodated through model finetuning (Zhou et al., 2025). We employ our VLM-friendly DSL in a similar manner.

## 3 DOMAIN-SPECIFIC LANGUAGE

MetaDSL uses a hierarchical, compositional approach centered around the *Tile* object, which is our version of an FBV (Section 2). As shown in Figure 1(c,d), a *Tile* describes a small representative unit of the design; then a *Pattern* propagates the tile into a space-filling *Structure*. Unlike previous works, MetaDSL eschews global coordinates in favor of local, normalized coordinates. These coordinates are expressed in relation to named entities of an abstract convex polytope (ACP) – such as the dimensionless cuboid in Figure 1(a,b) – which preserves semantic meaning and design intent. Any volumetric shape defined relative to an ACP is known as a *Volume* (Figure 1(b)). A Volume becomes concrete only after its ACP has been embedded – and thus, transformed into a Tile – by assigning a global coordinate to each corner of the ACP. These abstractions (Volume, Tile, Pattern, and Structure) form the basis of MetaDSL's expressive power by enabling modularity, reuse, verifiable construction, and our ability to leverage the strengths of existing metamaterial representations.

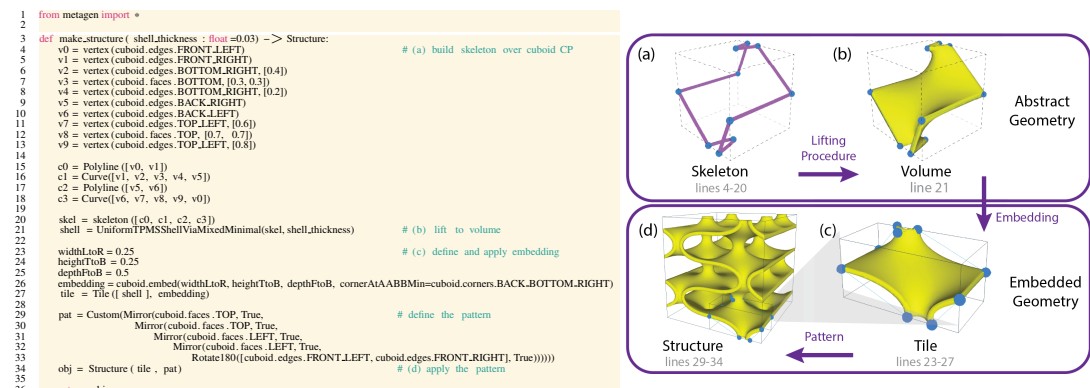

```
1   from metagen import *
2
3   def make_structure ( shell_thickness : float =0.03) −> Structure:
4       v0 = vertex (cuboid.edges.FRONT_LEFT)                              # (a) build skeleton over cuboid CP
5       v1 = vertex (cuboid.edges.FRONT_RIGHT)
6       v2 = vertex (cuboid.edges.BOTTOM_RIGHT, [0.4])
7       v3 = vertex (cuboid.faces.BOTTOM, [0.3, 0.3])
8       v4 = vertex (cuboid.edges.BOTTOM_RIGHT, [0.2])
9       v5 = vertex (cuboid.edges.BACK_RIGHT)
10      v6 = vertex (cuboid.edges.BACK_LEFT)
11      v7 = vertex (cuboid.edges.TOP_LEFT, [0.6])
12      v8 = vertex (cuboid.faces.TOP, [0.7, 0.7])
13      v9 = vertex (cuboid.edges.TOP_LEFT, [0.8])
14
15      c0 = Polyline ([v0, v1])
16      c1 = Curve([v1, v2, v3, v4, v5])
17      c2 = Polyline ([v5, v6])
18      c3 = Curve([v6, v7, v8, v9, v0])
19
20      skel = skeleton ([c0, c1, c2, c3])                                # (b) lift to volume
21      shell = UniformTPMSShellViaMixedMinimal(skel, shell_thickness)
22
23      widthLtoR = 0.25                                                  # (c) define and apply embedding
24      heightTtoB = 0.25
25      depthFtoB = 0.5
26      embedding = cuboid.embed(widthLtoR, heightTtoB, depthFtoB, cornerAtAABBMin=cuboid.corners.BACK_BOTTOM_RIGHT)
27      tile = Tile ([ shell ], embedding)
28
29      pat = Custom(Mirror(cuboid.faces.TOP, True,                        # define the pattern
30                      Mirror(cuboid.faces.TOP, True,
31                          Mirror(cuboid.faces.LEFT, True,
32                              Mirror(cuboid.faces.LEFT, True,
33                                  Rotate180([cuboid.edges.FRONT_LEFT, cuboid.edges.FRONT_RIGHT], True))))))
34      obj = Structure ( tile , pat)                                     # (d) apply the pattern
35
36      return obj
```

Figure 1: **(Left)** A MetaDSL program with skeleton-based Volume creation, and **(Right)** its construction stages: **(a)** build a Skeleton relative to a `cuboid` abstract convex polytope (ACP); **(b)** specify the procedure that lifts the Skeleton to a Volume; **(c)** create a Tile by embedding the ACP in $\mathbb{R}^3$ and executing the Volume procedure; and finally, **(d)** apply the specified Pattern to create a Structure.

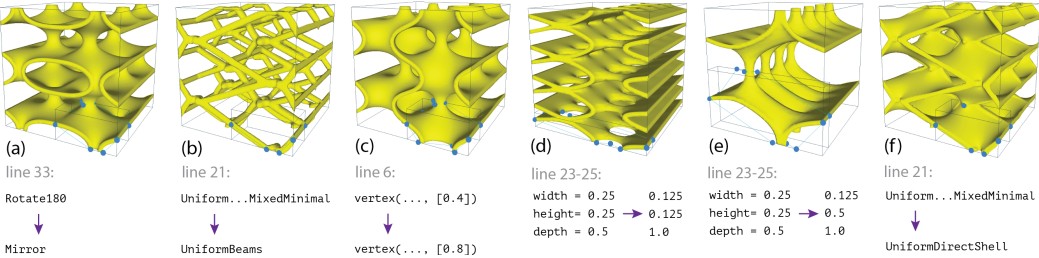

| (a) | (b) | (c) | (d) | (e) | (f) |
|---|---|---|---|---|---|
| line 33: | line 21: | line 6: | line 23-25: | line 23-25: | line 21: |
| Rotate180 | Uniform...MixedMinimal | vertex(..., [0.4]) | width = 0.25 → 0.125 | width = 0.25 → 0.125 | Uniform...MixedMinimal |
| ↓ | ↓ | ↓ | height= 0.25 → 0.125 | height= 0.25 → 0.5 | ↓ |
| Mirror | UniformBeams | vertex(..., [0.8]) | depth = 0.5 → 1.0 | depth = 0.5 → 1.0 | UniformDirectShell |

Figure 2: We illustrate the expressive power of MetaDSL by showing six different structures, each stemming from the program shown in Figure 1(a). Each one is produced by changing a single aspect of the original program, as detailed below each structure.

## 3.1 DSL STRUCTURE

A *Tile* is a finite, $\mathbb{R}^3$-embedded convex polytope (CP) that contains a volumetric metamaterial architecture. MetaDSL currently supports `Cuboid`, `Tet`, and `TriPrism` CPs. Each Tile maintains type information about its content – e.g., the architecture's connectivity and its incidence on the CP boundaries – that can be queried by downstream operators to determine validity and/or compatibility.

A *Pattern* is a sequence of spatial operations that compose with a Tile to promote it to a space-filling object (Structure). To promote modularity and reuse, patterns are defined relative to ACP topology (e.g. mirroring across the `TOP` face of a cuboid). MetaDSL currently implements 4 primitive patterns (`Mirror`, `Rotate180`, `Translate`, and `Identity`), a `Custom` pattern for custom compositions, and 3 aliases for recurring compositions (`TetFullMirror`, `TetPrismFullMirror`, and `CuboidFullMirror`, which replicate common 3D space groups (Adams & Orbanz, 2023; Hahn, 2002)). Patterns may be known sequences (e.g., apply two mirrors) or conditional procedures (e.g., mirror until a unit cube is filled). Patterns may also specify preconditions, such as the Tile's dihedral angles or aspect ratios, to be validated before application. Preconditions on the Tile's type annotations can also be used to ensure that architectural connectivity and continuity will be maintained.

*Structures* are space-filling objects given by patterned Tile(s). They constitute a complete procedural representation of a metamaterial. Structures can also be combined using standard CSG operations (`Union`, `Subtract`, and `Intersect`) to accommodate composite metamaterials with mixed scales, interpenetrating lattices (White et al., 2021), or multi-tile patterns Figure 3(e,f).

*Volumes* are abstract 3D shapes defined relative to an ACP. Specifically, a Volume is a function that takes a polytope Embedding (mapping each ACP corner to a position in $R^3$) and then returns

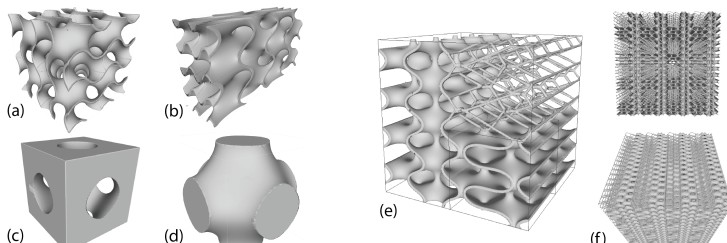

Figure 3: MetaDSL captures structures beyond those of the closest previous work, ProcMeta. **(a-d)** Our SDF-based Volumes permit common implicit structures: gyroid sheet TPMS over uniform (a) and non-uniform (b) tiles, and the Schwarz P exo-network (c) and endo-network (d). **(e,f)**: Multi-tile patterns interleave compatible structures from Figure 1(e) and Figure 2(b,c) to form a larger triply-periodic cell; (e) shows one multi-tile unit, while (f) shows two views of a 4x4x4 block.

a concrete 3D shape. The power of this abstraction is shown in Figure 2(a,d,e): despite using an identical Volume, each Tile embedding yields a distinct architecture. Thanks to our Volume abstraction, MetaDSL Tiles and Patterns are agnostic to the underlying shape description – i.e., Volumes can be described using any of the myriad approaches in Section 2. Our initial release includes two techniques: Skeleton-based construction and implicit definition via SDFs.

Our Skeleton-based method mirrors ProcMeta (Makatura et al., 2023). As shown in Figure 1, a `Skeleton` constitutes a 1D edge network of `Polylines` and `Curves`. Both edge types connect vertices that are defined relative to an ACP – e.g. `vertex(cuboid.edges.TOP_LEFT)` is at the center of the top-left edge of a cuboid ACP. Complete `Skeletons` are elevated to Volumes via Lifting Procedures. MetaDSL's current lifting functions generate beam structures from arbitrary edge networks (`UniformBeams` and `SpatiallyVaryingBeams`), shells from closed loop skeletons (`UniformDirectShell`, `UniformTPMSShellViaConjugation`, and `UniformTPMSShellViaMixedMinimal`), and spheres from singleton vertices (`Sphere`). To ensure compatibility, each entity (vertex, edge, skeleton) has type data that can be queried by downstream operators. This yields a range of beam, shell, and CSG-like structures (Figures 2 and 4).

In our SDF-based approach, the Volume is given by an implicit function over $\mathbb{R}^3$, expressed using arbitrary Python code. First, a `CartesianVolume` (CV) object anchors a Cartesian grid to the entities of an ACP. Then, a Lifting Procedure promotes the implicit function to a Volume by applying marching cubes to extract a mesh, then using the CV to transform the mesh into a collection of ACP-referenced entities. Once the Volume is defined, the MetaDSL framework applies as usual; this allows us to seamlessly integrate geometries that were previously out-of-scope (Figure 3(a-d)).

Together, MetaDSL's four abstractions (Volume, Tile, Pattern, Structure) form a uniquely powerful language for metamaterials. Each layer is independent and polymorphic, which promotes re-use and reconfiguration: Volumes embed in many Tiles (c.f. Figure 2(a,d,e)); Tiles propagate by many Patterns (c.f. Figure 2(a), Figure 1(d)); and Patterns apply to many Tiles (c.f. Figure 2(b,c,f)). Through type-enabled compatibility checks, MetaDSL allows complex composition with verifiable outcomes. MetaDSL's approach for Volume construction also allows us to choose the "right" description for each architecture, which ensures compactness, precision, and interpretability while preserving downstream compatibility. Meanwhile, MetaDSL's local coordinates simplify spatial reasoning for both humans and VLMs, by (1) using common English identifiers for topological entities, and (2) describing continuous relationships (like the position along a particular edge) in a normalized [0,1] range.

### 3.2 IMPLEMENTATION

We implemented MetaDSL as an embedded DSL in Python, which provides programmatic constructs (e.g., loops, functions, parameters) and semantic annotation (e.g., comments, variable/function names). Python's familiar, human-inspired syntax also smooths the transition between natural language and geometry. This allows MetaDSL to capture precise, complete metamaterial descriptions via programs that are compact, modular, verifiable, extensible, and suitable for humans and computational tools. The complete documentation of MetaDSL is in Section H.2, with additional discussion in Section C.

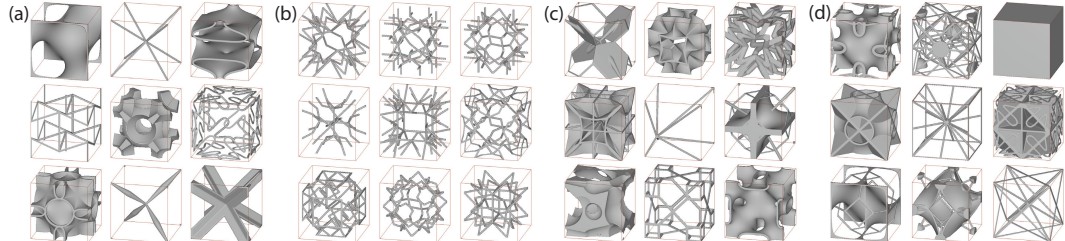

Figure 4: Assortment of metamaterials in MetaDB, illustrating four creation modes: (a) hand-authored seeds, (b) generated models, (c) type-enabled mutations, and (d) LLM-augmented hybrids.

Since MetaDSL captures *procedures* rather than *geometry*, each program's architecture can be realized using various geometry kernels and formats (voxel/tet/triangle meshes, SDFs, etc.). For a given backend, our `Structure` object is converted into suitable kernel instructions using a translation layer. Our first supported backend was ProcMeta, which is why all materials in the initial MetaDB release reside in a unit cube (as ProcMeta excludes geometry beyond that scope). However, MetaDSL is not bound by these limitations – using an alternate geometry kernel (Section C.5), MetaDSL already shows support for non-unit cubes (Figure 3(b)).

## 4 DATABASE GENERATION

We constructed a large, multifaceted database (**MetaDB**) of $153,263$ MetaDSL programs along with watertight geometry, renderings, simulated properties, and provenance metadata. All MetaDB materials are *validated* with geometric and physical checks (Section D.6).

### 4.1 MATERIAL GENERATION

The core challenge behind MetaDB is constructing a large, diverse, unified collection of high-quality programs. We tackled this by hand-authoring 50 parametric MetaDSL programs and one *program generator* – which itself yields over 1200 parametrized truss families – based on existing metamaterial literature. After generation and parameter sampling, this yields 36,997 *expert* materials. We *augment* these using two strategies, hybridization and mutation, that more than quadruple the database. We demonstrate the efficacy of these strategies in Section 6.

*Hybridization* (crossover), is motivated by works that offer extreme mechanical properties by super-imposing common structures: trusses with woven beams (Surjadi et al., 2025), nested trusses (Boda et al., 2025), TPMS shells with planar shells (Chen et al., 2019), and trusses with solids (White et al., 2021). We emulate this process by prompting an LLM with pairs or triplets of parent programs, then requesting hybrid code. Our prompting strategy (detailed in Section D.3) follows insights from recent works in LLM-mediated program search (Li et al., 2025; Romera-Paredes et al., 2024).

*Mutation* uses MetaDSL types to analyze programs and apply targeted edits: swapping compatible lifting functions, exchanging Curves and Polylines, or perturbing vertex positions and volume thicknesses (as in Figure 2). These variations stem from works such as Akbari et al. (2022), which posits beam approximations of TPMS shells.

### 4.2 AUXILIARY DATA

Each MetaDSL program is stored alongside a watertight `.obj` mesh, a $100^3$ voxelization, greyscale rendered orthographic images from the front, top, right, and front-top-right viewpoints, and simulated material properties. Our simulation performs periodic homogenisation over the voxelization, using a base material with $E = 1$, $\rho = 1$, $\nu = 0.45$. We store the $6 \times 6$ stiffness matrix $C$ and 18 derived scalars: 6 global metrics—Young's modulus $E$, shear modulus $G$, Poisson ratio $\nu$, bulk modulus $K$, anisotropy $A$, volume fraction $V$—plus directional values for $E$ (3), $G$ (3), and $\nu$ (6). Meshes, voxelizations, and simulations are computed using the geometry kernel and simulator from (Makatura et al., 2023). MetaDB programs begin with a parseable YAML header that stores provenance information: references to literature, parent materials and/or generators in MetaDB along

with any augmentation strategies and parameters used to produce this material. This data forms a provenance graph that spans MetaDB and connects it with the metamaterial literature.

## 5 BENCHMARK CURATION

MetaDB allows us to derive metamaterial benchmarks (**MetaBench**) over a set of *task categories*. To enable flexible interactions, each task category may contain multiple *task types* for different available inputs. This section describes the current MetaBench, but the procedures outlined below also serve as a framework for benchmark generation and adaptation. Thus, future iterations of MetaBench could explore the impact of different task categories, inputs/outputs, or query formulations.

### 5.1 METABENCH TASK TYPES

Presently, MetaBench probes 3 task categories – reconstruction, material understanding, and inverse design, as described in Section 1 – via 12 task types. Our category set supports general VLM-assisted design, as each one corresponds to a common stage of iterative design: prototyping, evaluation and guided concept development (Ivanova et al., 2024). Reconstruction and material understanding also serve as subproblems for inverse design, which requires constant translation among performance, geometry, and geometry representation. The relevant information for each task type is as follows.

**Reconstruction**  Given $n \in \{1, \ldots, 4\}$ orthographic images, the desired output is a metamaterial whose rendered geometry matches the target. Because every model has four views (Section 4.2), each model contributes $\binom{4}{n}$ examples of the $n$-view task type. Predictions are scored by chamfer distance (CD) and intersection-over-union (IoU) over voxelized unit cells (used for robust detection), as well as validity (percentage of designs that pass MetaDB material validity checks).

**Material understanding**  Given a structure description, the desired output predicts six global properties: Young's modulus $E$, shear modulus $G$, bulk modulus $K$, Poisson ratio $\nu$, anisotropy $A$, and volume fraction $V$. Values are rounded to two significant figures. Our benchmark supports two task types: *multiview_and_code* (four images + DSL code) and *single_image* (one image). Predictions are measured by absolute normalized[1] error, averaged over the predicted properties.

**Inverse design**  Given a target property profile, the desired output is a metamaterial whose simulated properties satisfy the profile. We support six task types: each length-$n$ task requests $n \in \{1, \ldots, 6\}$ property targets. Targets may be exact values, ranges, or upper/lower bounds—e.g., "auxetic ($\nu < 0$)" or "volume fraction $V \approx 0.6$." To construct target profiles from a model, we (1) sample $n$ active properties from the model, (2) choose bounds for each, and (3) render a natural-language prompt using a grammar conditioned on each property's part-of-speech tag (adjective, verb, etc.). This process is detailed in Section F.2. Both the prompt and the underlying numeric targets are stored, so users can rephrase questions or bypass NLP entirely. Predictions are measured by absolute normalized error, averaged over the target properties. We also measure validity and, for tuned models, novelty (percentage of designs *distinct from* the training set, measured by voxelizations).

### 5.2 PROCEDURE FOR TASK-BASED DATASET CONSTRUCTION

First, we designate a pool of *active* MetaDB entries and partition them into train, validation, and test splits that remain fixed for all tasks. Given an entry and a task type, we gather all data used for query/response (ground truth) construction and prediction evaluation. We also form the query/response pairs using category-specific prompt templates (Sections H.3 to H.5). This information (query, response, and supporting data) is then stored in a model-agnostic intermediate format (Section F.1). Beyond easy model adaptation, this format allows researchers to reframe prompts without regenerating or deviating from the core content of the inquiry. It also allows alternate input/output representations – e.g., inverse design of voxel grids rather than MetaDSL – so long as the updated choices meet the preconditions of the relevant evaluation metric.

---

[1] Properties are linearly scaled so the dataset range is [0,1].

Table 1: Using MetaBench to evaluate various models' performance. LLaVA is not reported because it failed to produce any valid output. See Figures 14 to 16 for qualitative evaluation.

| | Inverse Design | | | Material Understanding | Reconstruction | | |
| | Error ↓ | Valid ↑ | Novel | Error ↓ | CD ↓ | IoU ↑ | Valid ↑ |
|---|---|---|---|---|---|---|---|
| MetaAssist (LLaVA) | .021 ± .002 | **98.1%** | 49% | .030 ± .004 | **.032 ± .001** | **.493 ± .008** | 94.2% |
| MetaAssist (Nova) | **.018 ± .002** | 89.5% | 76% | **.021 ± .003** | .035 ± .001 | .449 ± .007 | **97.5%** |
| Nova | .056 ± .023 | 2.6% | – | .199 ± .005 | .119 ± .003 | .051 ± .003 | 19.3% |
| OpenAI O3 | .028 ± .004 | 37.3% | – | .077 ± .005 | .053 ± .001 | .147 ± .004 | 54.6% |
| Constant (Mean) | – | – | – | .072 ± .007 | – | – | – |
| Random (Normal) | – | – | – | .122 ± .007 | – | – | – |
| Random (Uniform) | – | – | – | .414 ± .010 | – | – | – |

# 6 RESULTS

## 6.1 LANGUAGE DESIGN & DATABASE CREATION

By collating knowledge and strategies from across existing literature (challenge C3), MetaDB amasses one of the largest metamaterial databases ever collected, comprising $153,263$ materials. Expert designs comprise $24\%$ (36,997 entries) of the database; this includes 1,588 variations of 50 hand-authored programs, 1,205 generations, and 34,204 generation parameter variations. We also introduce 12,029 hybrids and 141,234 mutations. Our programs offer exceptional geometric fidelity and diversity (Figures 1 to 4). As shown in the inset plot, hybridization and mutation also drastically improve the achievable property range (gamut) and sampling density of MetaDB. Along individual property axes, the augmented gamut includes a near-doubling in anisotropies and quadrupling of some directional Poisson ratios.

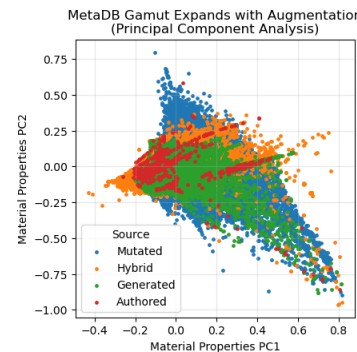

Our experiments also highlight the advantages of DSL representations for dataset augmentation. General mutation is possible because we can analyze type compatibility of operators, enabling valid updates well beyond simple parameter variation. Hybridization constructs interesting, valid materials by leveraging VLMs' existing familiarity with Python (MetaDSL's host language). In a hybridization experiment comparing the use of JSON serialized ProcMeta graphs and MetaDSL programs, the latter produces valid materials 39% more frequently with 45% fewer tokens, allowing for significantly higher augmentation throughput. This validates MetaDSL as a highly navigable design space for metamaterials, and positions DSLs as an ideal medium for VLM exploration.

## 6.2 BENCHMARKING

For the following experiments, MetaBench is sampled from the core material set containing 13,282 authored, generated, and hybrid models. We randomly split these models into 500 test, 50 validation, and 12,732 training materials, and generated benchmark tasks for each as described in Section 5.

**Experimental Setup** We used MetaBench to measure baseline performance of VLMs as metamaterial design assistants. We measured 3 zero-shot baselines: a small, open-source VLM `LLama3-LLavA-Next-8b` (LLaVA) (Li et al., 2024), a large commercial VLM `amazon.nova-lite-v1:0:300k` (Nova) ((Amazon AGI), 2024), and a large commercial chain-of-thought reasoning model (OpenAI o3) (OpenAI, 2025). We also used the MetaBench omnitask training set to apply supervised fine tuning (SFT) atop LLaVA and Nova (details in Section G). We call the resulting models MetaAssist (LLaVA) and MetaAssist (Nova).

All experiments and tasks share a common system prompt in both training and inference (Section H.1) containing the MetaDSL API documentation (Section H.2), a glossary of our naming conventions (e.g. E is the Voigt-Reuss-Hill Young's modulus), and a one-sentence description of each task category.

Table 1 shows the performance for our two MetaAssist models, three zero-shot VLM baselines, and constant/random baselines for material understanding, as measured by the metrics in Section 5.1.

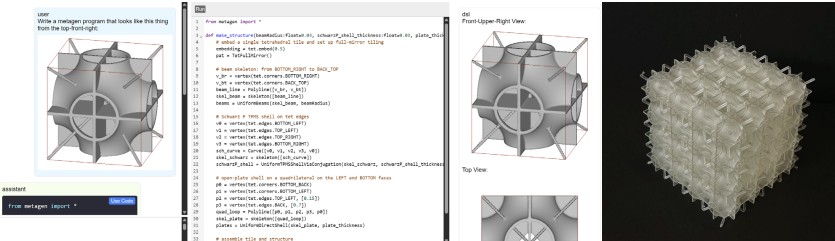

Figure 5: Reconstruction: (Left) Generating a metamaterial program from an input image enables incorporating designs from literature, sketches, and nature. (Right) 3D printed design.

**Experimental Insights**  Our experiments demonstrate that off-the-shelf VLMs perform poorly across all tasks and metrics. This holds across all three of our untrained models, suggesting that performance is independent of model scale. As such, we conclude that at least some finetuning is required for VLMs to support DSLs targeted at niche or technical domains.

However, even after simple SFT, both MetaAssist variants achieved high material validity rates in generative tasks (reconstruction and inverse design), and low errors. Qualitative understanding of these errors is illustrated by results galleries in Section E. This suggests that both large (MetaAssist(Nova)) and small (MetaAssist(LLaVA)) models are capable metamaterial design assistants. However, MetaAssist(Nova) appears to have an increased capacity for generalization, as it produces novel material designs $1.5\times$ as often. When the results are broken down by task type (Section E.2), MetaAssist(Nova) also proves itself better equipped to parse and leverage large amounts of input information. This is particularly clear across the material understanding task types: MetaAssist(Nova) shows a 30% reduction in error from the 1-view task (0.024) to the 4-view-plus-code task (0.017), while MetaAssist(LLaVA) shows an 18% increase ($0.028 \rightarrow 0.033$). Perhaps surprisingly, MetaAssist(LLaVA) occasionally outperforms the significantly larger MetaAssist(Nova) model. This is likely to due a Nova-specific post-training step that aims to maintain general task capabilities (Section I).

Both MetaAssist models' strong performance on inverse design also represents a significant contribution, by demonstrating VLMs' ability to traverse not one but *two* complex domain mappings (code↔geometry↔performance) across three separate modalities. It also constitutes the first general inverse-design approach that is mesh- and simulation-free at inference time. This will improve the time and complexity of individual design iterations compared to traditional techniques like TO, which does expensive simulation in a loop. For improved accessibility, we also operate over lower-burden inputs (based on natural language, with approximate target bounds and several objectives). Due to these different goals, TO and other inverse design methods are not suitable for direct comparison.

**Ablation**  To analyze the effect of generalist versus specialist models, we complemented each of the two omnitask MetaAssist models with a set of 3 single-task counterparts, each finetuned over a particular task type (4-view reconstruction, 4-target inverse design, and multi-view-plus-code material understanding). As shown in Section E.1.2, multi-task models outperform single-task models across nearly all models and tasks. This demonstrates that the tasks are complementary and mutually beneficial. It also supports finetuning over both sub- and super-tasks, because supertasks simultaneously benefit from and evaluate the quality of the models' learned subtask embeddings.

## 6.3 INTERACTIVE CASE STUDIES

To illustrate the potential of a metamaterial design copilot, we also built a browser-based chat and MetaDSL editor to explore design interactions with MetaAssist (Nova) – selected due to its large context window and stronger conversational abilities. We present a scenario that generates a material from an input image, as one might do when (re)creating naturally-occurring structures, materials pictured in literature, or sketched design concepts. Figure 5 shows this functionality with a material from the MetaBench test set; even with our conversational (rather than structured) request, we were still able to obtain and fabricate a perfect reconstruction.

The second scenario is iterative inverse design (Figure 6), in which we specify a set of target property bounds, and the model generates a metamaterial that satisfies them (as verified with our simulator).

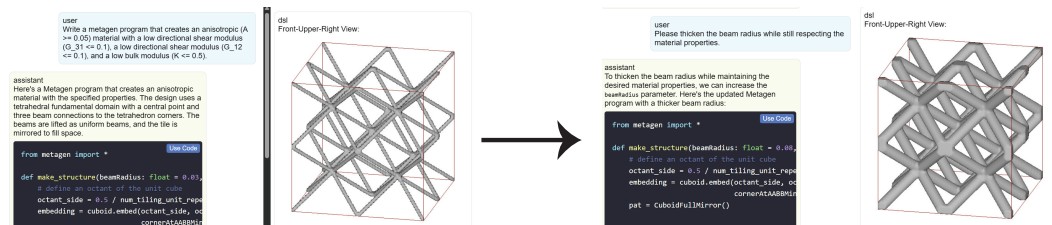

Figure 6: Iterative Inverse Design: Designers can specify desired target properties, and these preferences and constraints can be considered throughout multiple design iterations.

But design is always iterative, and seeing one design can spark new criteria and objectives. In this case we wanted a thicker structure that still conformed to our original input; again, the model was able to update the design within target parameters. This illustrates LLMs' powerful ability to remember and carry through *design context*, allowing for assistance across multiple design iterations.

## 7 DISCUSSION, LIMITATIONS, & FUTURE WORK

Metamaterial design is a high-impact, multimodal domain for which accessible, general VLM-assisted design tools promise enormous value. Metamaterial benchmarks may also drive VLM advancement, as they offer a meaningful, challenging testbed for e.g. interactive workflows, reinforcement learning (RL), equivariant networks, and verified program synthesis. To facilitate this symbiotic exchange, MetaDSL and MetaDB provide a common, traceable descriptor that both communities can adopt. As researchers contribute new designs, the database will grow organically, giving AI practitioners richer training data while delivering state-of-the-art design assistants to materials scientists.

Our work provides a comprehensive framework toward these goals, with many avenues for improvement. MetaAssist was deliberately restricted to simple supervised fine tuning to provide a bedrock baseline for our new task. Future works may incorporate RAG to read papers and retrieve patterns, CoT reasoning to connect design intent to property profiles, or RL with curriculum learning to generalize to novel inverse design profiles. Future works may also leverage MetaDSL's retargetability (Section B): faster, more flexible kernels would enable larger interactive workflows, simulation-in-the-loop optimization, even-wider dataset scales, and non-cubic and/or aperiodic tilings.

Our knowledge base also has ample opportunities for growth and refinement. For example, a recent survey identified 5 "typical" metamaterial representations (Lee et al., 2024); MetaDSL's initial release captures structures from 4 of them. The fifth class – voxel grids – could be captured directly (via a CV-based lifting function that takes a binary grid), or indirectly (by fitting them with skeleton-based templates (Chen et al., 2018)). MetaDB could also be expanded by design generators (Sun et al., 2023a; Liu et al., 2022; Abu-Mualla & Huang, 2024; Makatura et al., 2023) and diversity-guided synthesis strategies (Chan et al., 2020). Toward better understanding, our program's explicit semantic structure may also facilitate taxonomy construction (Zok et al., 2016). With broad participation, MetaDB could become the primary resource for tracking metamaterial lineages, structure–property relationships, and mechanistic insights—paralleling the role ImageNet played in computer vision.

## 8 CONCLUSION

We introduced **MetaGen**, a unified ecosystem for VLM-assisted metamaterial design that combines (i) *MetaDSL*, a compact yet expressive domain-specific language that is uniquely suited for human and VLM use; (ii) *MetaDB*, a 150 000-entry database with paired geometry, renderings, and physics; and (iii) *MetaBench*, a task-oriented benchmark that probes reconstruction, material understanding, and inverse design. Our baseline experiments illustrate that VLMs offer promising performance for multi-modal translation and design generation, even for tasks that require multiple functional maps between characteristically different modalities. Moreover, we provide a holistic vision for accelerated, symbiotic research at the intersection of machine learning and architected materials. With MetaGen offering both a challenging domain for VLM development and a practical toolkit for materials scientists, our paper lays the foundation to bring this vision to life.

ETHICS STATEMENT

To facilitate responsible VLM-assisted design, any deployment of MetaAssist models should include safety guardrails that mitigate the potential for errors or misguided application. This deserves particular attention in a domain like metamaterials, which is complex and rapidly evolving, yet targeting applications in which failures may be catastrophic. As such, results must be validated and communication measured. Our format already takes small strides toward transparency by maintaining detailed provenance records, and releasing our artifacts along with the pipelines used to generate them. Moving forward, it would be prudent to include additional safeguards such as automated validity checks, uncertainty estimates, safety factors, and access to high-fidelity simulators to reduce the risk of erroneous or unsafe designs.

REPRODUCIBILITY STATEMENT

All data and code will be made publicly available, under permissive open source licensing. Contribution guidelines will be determined after release, to ensure that the policies are sufficiently flexible to accommodate the community needs and preferences regarding use cases, extensions, etc.

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

## A  APPENDIX

## B  ECOSYSTEM DESIGN

The four components of the MetaGen ecosystem work together to achieve our design goals. We outline these goals and the design and organization decisions that achieve them here:

- MetaDB
    - Design Goals: Collect existing knowledge in a reconfigurable, reusable, and task independent manner
    - Organization
        * Primary Elements: Material Definitions; Provenance
        * Derived Elements: Geometry; Computed Properties
- MetaBench
    - Design Goals:
    - Organization:
        * Primary Elements: Structured Task Definitions; Target Data; References, Evaluation Procedures
        * Derived Elements: Query Strings; Example Responses
- MetaDSL
    - Design Goals: Eventual Comprehensiveness via Extensibility; Supports Hybrid Structures Easily; Ease of Use
    - Design Decisions: Extensible Embedded Python DSL for extensibility and Ease-of-Us; Separation of Front-End Language from Geometry Kernel

- MetaAssist
  - Design Goals: Usable for general engineers; single interface across design silos; possibility of integrating unstructured data (literature, sketches, etc.)
  - Elements: Interactive Interface; Trained Baseline Models

Each component supports the others, as illustrated in Figure 7

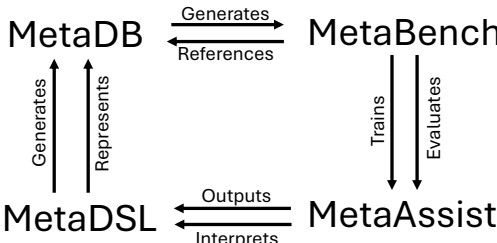

Figure 7: Relationships between MetaGen ecosystem components.

### B.1 Ecosystem Development and Insights

The elements of this ecosystem were developed in concert with one another, going through 3 major iterations before arriving at their current state. MetaDSL was at the heart of each iteration, as the representation has a direct impact on the efficacy of the other three components:

- MetaDB needs a representation that captures diverse structures, but also offers robust pathways for scalable (and, in this case, VLM-driven) structure generation, hybridization, mutation, sampling, etc.
- MetaBench can only be used for training and evaluation if it is built atop a large, diverse database.
- MetaAssist relies on a strong training corpus from MetaBench. MetaAssist also hinges on the intelligibility of the representation, and the model's ability to interpret, generate, and modify programs according to user input.

We defer the language-specific development details to Section C.6.

Outside the scope of the DSL, we also found that dataset management and curation posed a major hurdle. We improved diversity by continuously mining metamaterial literature for additional seed program designs. We expressed these seed programs as-parametrically-as-possible to allow for expert-driven sampling. As we scaled the dataset, we also realized that it would be critical to keep track of the programs' sources and relationship to one another. This information is especially useful for navigation, contextualization and diversity management, particularly as the database grows in response to community effort. To manage this, we introduced a formalized provenance system for MetaDB.

## C    MetaDSL

### C.1    Additional Implementation Details

We implemented the core functionality of MetaDSL (version 1.1.0) with two goals in mind. First, we wanted full support for the metamaterials that were expressible in our geometry kernel, ProcMeta. Second, we wanted our infrastructure to easily permit extensions in the future without invalidating existing programs. We detail the current state of each feature category in our language: convex polytopes, skeletons, lifting procedures, tiles, and patterns. For a full API description of the accessible functions, please refer to Section H.2. Figure 8 shows an overview of the compiler architecture.

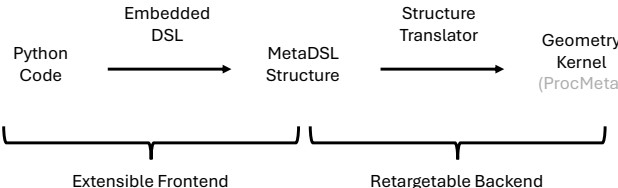

Figure 8: Overview of MetaDSL's implementation. MetaDSL programs are written in an embedded Python DSL frontend to allow for ease of use and extensibility. These structures are compiled into a structured intermediate representation, and a backend Translator converts these structures into geometry kernel instructions. In our implementation we used the geometry kernel from ProcMeta Makatura et al. (2023). By separating the front-end representation from the backend geometry kernel, MetaDSL is flexible to both be extended in its frontend representation, and retargettable to different geometry backends for new applications, while keeping a compatible material representation.

**Convex Polytopes (CP)**   Currently, all of our programs make use of three pre-defined CPs (as inspired by ProcMeta): `cuboid`, `triPrism` and `tet`. The infrastructure to define custom convex polytopes exists, and most operators up to and including Tiles should generalize to such CPs. However, the patterning operations would need to be generalized before being able to operate on arbitrary CPs.

**Skeletons**   Then, a *skeleton* is constructed via a set of vertices and edges that are positioned relative to a common CP. Each vertex is positioned on a particular CP entity (corner, edge, face, interior). Each CP entity is accessed via a semantically meaningful alias, permitting calls such as e.g. `vertex(cuboid.edges.BACK_LEFT)`. The `vertex` call also optionally takes a list $\vec{t}$ of interpolation values used to position the vertex within the entity. If $\vec{t}$ is omitted, the returned point will be at the entity's midpoint (edge) or centroid (face/interior). Presently, corners ignore weights (since they cannot be moved); edges use linear interpolation; and faces use barycentric coordinates if they contain 3 vertices or bilinear interpolation for quads. If a CP with different polygonal faces (e.g. pentagons) were implemented, an appropriate lower-dimensional vertex positioning specification would need to be devised. Internally, the vertices are stored using weights over a full list of the CP corners, so additional specification interfaces can easily be defined.

An ordered list of vertices can then be strung together into simple (non-branching, self-intersection-free) open or closed paths via the `Polyline` or `Curve` commands. Each edge contained in a path infers and maintains information about its incidence on the CP – including whether it is contained within a face, through the CP volume, coincident with a CP edge, etc. This is very useful when determining lifting function compatibility, as some procedures can only be applied when e.g. every path edge is contained within a CP face.

Then, a `skeleton` is used to combine a set of vertices or polylines/curves into a larger, more complex element, over which additional organizational information is computed. Skeletons infer the connected components formed by the inputs, then categorize them based on their topology. Thus, a skeleton may be labeled as a simple closed loop, even if the input is a set of open paths. Again, these insights are critical for determining the skeleton's compatibility with downstream operations, such as lifting procedures. We also included infrastructure for the skeletons to infer and track their total incidence on each entity of the reference CP, including the dimensionality (e.g. point or line) of an intersection – however, this feature is not fully implemented in the current MetaDSL version.

**Lifting Procedures**   *Lifting procedures* are used to transform the skeleton into a volumetric object. Simple procedures like `Spheres` instantiate a sphere of the given radius centered at each vertex in the skeleton. Similarly, `UniformBeams` instantiates a beam of the given thickness centered along each path of the input skeleton. The shell operators (`UniformDirectShell`, `UniformTPMSShellViaMixedMinimal`, and `UniformTPMSShellViaConjugation`) solve for a surface that spans the provided boundary curve before expanding the surface to the desired thickness. Our shell and beam procedures mimic those defined by ProcMeta, as they cover

```
972
973   from metagen import *
974   def make_structure ( shell_thickness =0.03) -> Structure:
975       v0 = vertex ( tet .edges. BOTTOM_LEFT)
          v1 = vertex ( tet .edges.TOP_LEFT)
976       v2 = vertex ( tet .edges.TOP_RIGHT)
977       v3 = vertex ( tet .edges. BOTTOM_RIGHT)
978
979       c0 = Curve([v0, v1, v2, v3, v0])
980
          skel = skeleton ([c0])
981       shell = UniformTPMSShellViaConjugation(skel, shell_thickness )
982
          embedding = tet .embed(0.5)
983        tile = Tile ([ shell ], embedding)
984       pat = TetFullMirror ()
          obj = Structure ( tile , pat )
985
986       return  obj
987
```

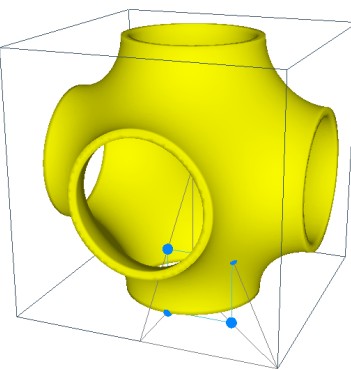

Figure 9: Example program and corresponding geometry for the Schwarz P structure.

a wide range of metamaterial classes and were already (by construction) natively supported by our geometry kernel. Our `Curve` and `Polyline` commands correspond to their smooth/non-smooth edge chains, respectively. Unlike the original, we chose to explicitly separate several operators that were previously lumped together, which clarified and minimized the number of exposed parameters for each call.

**Tiles**   To create an embedded, patternable tile, we provide a list of one or more lifted skeletons as input to the `Tile` operator. The tile operator also takes as input the embedding information, which will be used to embed the CP and, in turn, each vertex of the contained skeleton(s). To obtain the embedding information, each CP implements at least one `embed` function, which takes high level parameters such as the min/max position of the CP's AABB.

Because of constraints imposed by $ProcMeta$ – that these must form a partition of the unit cell – our code currently treats these CPs with some additional assumptions. Specifically, though the cuboid need not be a cube, it must have right angles everywhere, and edge lengths must be $1/2^k$ for some positive integer $k$; in practice, $k \in [1, ..4]$. The triPrism is assumed to be an isoceles triangle with a right angle. The tet similarly has a base that is an isoceles triangle with a right angle, and a fourth vertex that is located directly above one of the $45$ degree angles. These assumptions would ideally be relaxed in a future version of MetaDSL.

**Patterns**   Patterns are currently the most restricted feature of in the core release of MetaDSL, as we only implemented patterning operations that were realizable in ProcMeta– i.e., those that lead to translational units within a unit cube. The pattern operators were written in a way that allows for additional, extended tiling procedures. We prioritized mirrors, because they are sufficient to express a wide range of common metamaterial designs, and they are often used in generative metamaterial design schemes, as the connectivity requirements are simpler than most other operations. We also have limited support for other operations such as `Rotate180` and `Translate`, which can be used inside the `Custom` pattern specifier. Currently, these limited operations are only defined for specific transformations on cuboids. We look forward to an expanded MetaDSL that includes full support for these patterning operations, at least over the pre-built CPs that currently exist. In the long term, we envision a patterning system that extends well beyond this, to support large, potentially aperiodic or asymmetric tilings composed of one or more tiles with arbitrary CPs. This is a very difficult problem, and will itself present an interesting set of research directions, including how to intuitively specify these patterns and how to characterize their compatibility/validity.

### C.2   EXAMPLE PROGRAMS

Example program-structure pairs are listed in Figure 9 and Figure 10. Many additional models can be found in the accompanying data.

```
from metagen import *

def make_structure (beamRadius_narrow=0.03, beamRadius_wide=0.1) -> Structure:
    embed = cuboid.embed(0.5,  0.5,  0.5,
                        cornerAtAABBMin=cuboid.corners.FRONT_BOTTOM_LEFT)

    v0 = vertex (cuboid. corners .FRONT_BOTTOM_LEFT)
    v1 = vertex (cuboid. corners .BACK_TOP_RIGHT)
    p0 = Polyline ([v0, v1])

    skel  = skeleton ([p0])
    liftedSkel  = SpatiallyVaryingBeams(skel,  [[0,  beamRadius_narrow],
                                                [0.5,  beamRadius_wide],
                                                [1,  beamRadius_narrow]])

    tile  = Tile ([ liftedSkel ], embed)
    pat = Custom(Rotate180([cuboid.edges.BACK_RIGHT, cuboid.edges.BACK_LEFT], True,
                        Rotate180([cuboid.edges.TOP_RIGHT], True)))
    obj  = Structure ( tile , pat )

    return  obj
```

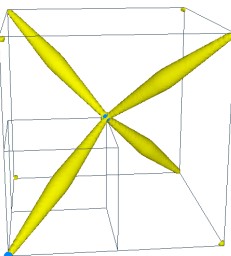

Figure 10: Example program and corresponding geometry for the pentamode structure.

## C.3 UNIQUENESS

MetaDSL programs are not naturally unique; there exist many ways to represent a given metamaterial structure. For example: the Schwarz P shown in Figure 9 is represented over a tetrahedral bounding volume representing $1/48$ of structure's the translational unit, but it could just as easily be conceived over a cuboid bounding volume representing $1/8$ of the translational unit, as shown in Figure 1 of Makatura et al. (2023). There is no clear "best" representation for metamaterials. We could select a minimal representation as the conventional descriptor, which would prioritize the tet-based Schwarz P over the cuboid-based one. However, it may be that the cuboid-based structure is more intuitive for people; since readability is a key consideration in human-facing interfaces, this ought to be considered when selecting standards.

Uniqueness is also difficult because there are often disparate representations that can lead to similar or identical structures. For example, when TPMS structures like the SchwarzP appear in literature, they are most commonly approximated using a simple trigonometric implicit function. We discuss an extended MetaDSL that can accommodate such implicit functions in Section C.5. These are not identical to our ProcMeta-inspired representation above, but they have considerable agreement, and are often considered equivalent in practice.

Moreover, since metamaterials are generally intended to tile $\mathbb{R}^3$, uniqueness can only be determined on the already-tiled scale. This is because there are infinitely many translational unit cubes that lead to the same tiled structure in $R^3$: the unit cell extracted from $[0]^3 - [1]^3$ is equally valid as the one from $[-0.5]^3 - [0.5]^3$, but they are likely to appear very different on their own.

Because of the ambiguities discussed above, MetaDSL makes no effort to encourage or enforce a canonical representation.* However, our provenance block can be fruitfully used to track relationships such as these – in many of the hand-authored examples, there is already a partial record of related structures that appear in the database. As MetaDSL and MetaDB evolve over time, it would be very interesting to track related representations and establish canonical norms.

* Although MetaDSL currently ignores uniqueness metrics, we do make some preliminary efforts to prevent excessive duplication in MetaDB. Specifically, we evaluate the voxelized representation of our non-expert materials (e.g., hybridizations, augmentations) and compare it to existing samples in the database. The new material candidate is only added if it is sufficiently different.

## C.4 METADSL VS. PROCMETA

As suggested by Section B.1 and the architecture diagram in Figure 8, MetaDSL is distinct from and strictly more general than ProcMeta, with a design philosophy all its own. Our approach was motivated by our early experiments with ProcMeta, which revealed a critical shortcoming: important information was represented implicitly in the ProcMeta GUI interface, and was entirely absent from

the ProcMeta graph representation. To make this information accessible to LLMs (and more easily accessible to humans), we implemented a programmatic interface, MetaDSL, that compiles to the same geometry kernel as ProcMeta, but provides several practical advantages (see Table 2).

Most importantly, MetaDSL introduces explicit, referenceable bounding volumes (BVs), which are critical for verifying and enforcing the preconditions of geometry operations. In the ProcMeta GUI, BVs exist only as non-referenceable visual aids; users must manually align coordinates, and no automated compatibility checks are possible. ProcMeta graphs omit BVs entirely. MetaDSL represents BVs through a CP abstraction, which enforces constraints by construction, enables type checking, and cleanly separates tile content from patterning, improving modularity and reconfigurability. These features align the representation more closely with the valid shape space, aiding both human designers and LLMs in producing valid, diverse structures. MetaDSL programs also make heavy use of programmatic features absent from ProcMeta graphs. Semantic variable names, comments (avg. 4/program), and parametric variables improve human interpretability and support natural-language reasoning for LLMs. Loops and helper functions are also common, appearing in 1,744 and 2,103 of the 13,284 core programs respectively. These features allow compact, self-consistent definitions that would be unwieldy if unrolled or inlined into a ProcMeta graph.

We tested LLM-based augmentation using ProcMeta JSON instead of MetaDSL. MetaDSL yielded: (1) higher code validity (75% vs. 54%), (2) more structurally focused reasoning rather than boilerplate handling, and (3) lower token usage (580 vs. 1,049 tokens on average for o4). Beyond these immediate benefits for LLM usage and dataset generation, our DSL interface also makes MetaDSL a more flexible platform from which to build further extensions, which facilitates its intended purpose as the seed of a wider community project.

## C.5    EXTENSIBILITY

The MetaDSL interface naturally generalizes to shape spaces that would be difficult to represent in ProcMeta's graph approach. For example, implicit functions are common in metamaterial design, but they would be cumbersome to represent in ProcMeta's graph. To show that MetaDSL could extend to such structures, we implement a separate, proof-of-concept pipeline that circumvents the ProcMeta backend.

**MetaDSL Extension**    First, we extend the MetaDSL language to accomodate implicit function-based skeleton generators and SDF-based lifting functions.    To do this, we introduced a `CartesianVolume` object, which anchors a Cartesian grid to the entities of $\Pi_{abs}$, such that local Cartesian coordinates can automatically be transformed into CP-referenced entities. This allows users to define and manipulate SDFs in a familiar workflow, while preserving MetaDSL's abstraction and validation capabilities. An example program is available in Figure 11.

**Alternate Geometry Kernel & Transpiler**    We implemented a simple backend that iteratively builds up a triangle mesh based on the operations (e.g., create vertex, create face, mirror) requested by a topologically-sorted input graph. Our kernel only supports inputs derived from MetaDSL Structures with fully-known geometry at the time of export. This excludes most structures that are derived from ProcMeta-inspired lifting functions (e.g., `UniformTPMSShellViaConjugation`), because the precise geometry is not known at the time of export; it is inferred by the ProcMeta kernel. As such, at this time, our simple kernel only supports SDF-based skeletons – e.g., those derived from `liftedSkelsFromSDF`. This could be expanded in the future.

To use our simple kernel to realize compatible MetaDSL structures, we wrote a new transpilation layer. This layer traverses the MetaDSL `Structure` object, and outputs a sequence of ordered operation nodes that match the interface given by our transpiler. This code is very similar to the transpiler required for ProcMeta.

Despite the simplicity of these extensions, they greatly expand the representative capacity of MetaDSL by incorporating the suite of implicit surfaces and other SDF based structures. Our simple, separate geometry kernel also relaxes one of the central limitations of ProcMeta: due to its simplicity, it does not need to make any assumptions about the form factor of the final structure. This permits translational units that reside in something other than a unit cube. Overall, this proof-of-concept demonstrates the capacity of MetaDSL to expand in a productive, non-breaking way.

```python
from metagen import *
from sdf import *
from tpms_helpers import *
from common_tpms import gyroid

def make_structure(isoval_min: float=-0.2, isoval_max: float=0.2, l: float=1.0) -> Structure:

    cv = CartesianVolume(    cuboid.corners.FRONT_BOTTOM_RIGHT,
                             cuboid.corners.FRONT_BOTTOM_LEFT, 1,
                             cuboid.corners.FRONT_TOP_RIGHT,    1,
                             cuboid.corners.BACK_BOTTOM_RIGHT, 1)

    shell_sdf = sheet_isosurface_pair(gyroid, isoval_min, isoval_max)
    shell = cv.liftedSkelsFromSDF(shell_sdf)
    print(f"Num ccs: {shell[0].skel.num_connected_components()}")
    print(f"Some cc on all faces: {shell[0].skel.is_some_cc_on_all_faces()}")

    # embedding and tiling
    side_len = 1.0
    embedding = cuboid.embed(0.5*side_len, side_len, 2*side_len)
    tile = Tile(shell, embedding)
    pat = Custom(Translate(cube.faces.FRONT, cube.faces.BACK, True,
                    Translate(cube.faces.BOTTOM, cube.faces.TOP, True,
                        Translate(cube.faces.LEFT, cube.faces.RIGHT, True))))
    return Structure(tile, pat)
```

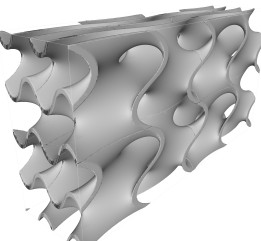

Figure 11: Example program and corresponding geometry for the implicit gyroid, with a stretched unit cell generated by embedding the Tile with a non-unit aspect ratio.

| | MetaDSL | ProcMeta |
|---|---|---|
| Compactness | **Shorter**, less boilerplate. Easier to read, less likely to exceed token limits | Longer, more boilerplate. Exceeds context of small, lightweight models. |
| Modules | **Highly reusable**. Patterns defined in composable chunks (eg TetMirror), independent of tile contents. Skeletons defined independent of embedding, easily scale to different Tiles. | **No support.** Limited reuse. Patterns can't exist independently; no pre-built Patterns. Absolute Skeletons, cannot easily be rescaled. |
| Relative vs. Absolute Positioning | Positions and transforms use **local coordinates** (i.e. [0,1]) wrt named entities (`cuboid.edges.TOP_LEFT`) in abstract polytopes. Robust for generation, clear design space bounds, more intuitive. | Positions and transforms use **absolute coordinates**. Easily misaligned, difficult to visualize without plotting. Unsuitable for VLMs, which struggle with computation/spatial tasks. |
| BV representation | **Explicit BV with named, referenceable entities.** Facilitates verifiable parametric design, e.g., vertex constrained to given BV edge. Allows type/error checking. | **Implicit or Absent BV**: drawn as a visual aid in the GUI, but not represented/preserved in the graph. Never referenceable. |
| Type/Error checking | **Type/incidence tracking to ensure compatibility** – e.g. conjugate TPMS require a closed loop where every edge lies in a BV face, and every BV face contains at least 1 loop edge. This is known from our representation and verified by downstream operations. Helps determine valid substitutions for mutations, even when large changes are proposed, leading to greater diversity. Critical for complex patterning, to determine compatibility of proposed-adjacent faces. | **None.** The burden of verification (for e.g. vertices on BV edges or edges in BV faces) is left to the user – infeasible for agentic design. Bad inputs crash ProcMeta with no explanation or suggested improvements. |
| Simplified Operations | **Abstractions simplify element creation**; e.g., Sphere() takes a center point and a radius, as one would expect. Easier for humans and LLMs. | Strict compliance with the given graph interface makes **some operations cumbersome**; e.g. for a sphere, thicken a 0-length edge chain over 2 co-located vertices |
| Semantic information | **Complete support.** Comments and meaningful variable names improve readability and admit metadata (provenance, parameter bounds) | No support. |
| Parameters | **Complete support.** Allows parametrized models and family generators. | **None.** Explicit positions etc. only. Variations defined as separate graphs. Difficult/impossible to infer constraints or design space from the graph description. |
| Loops, Functions | **Supports complex logic** that would be tedious to implement otherwise. Functions are especially useful for hybridization, as programs can be directly reused and/or rescaled. | **No support.** Each instance must be created/connected individually. Even hybridization is difficult, because subgraphs cannot be inserted directly – the identifier/references of each node must be updated. |

Table 2: Detailed differences between the interfaces for MetaDSL and ProcMeta.

### C.6    Language Development Process and Insights

As mentioned in Section B.1, our geometry representation went through 3 major stages.

In the first iteration, we represented metamaterials using ProcMeta graphs directly. This had several issues: it was not compact enough for the context windows of small, lightweight models; intuitiveness and editability suffered dramatically without the aid of a GUI editing tool; the graphs' use of absolute coordinates proved challenging for LLMs (which struggle with spatial reasoning); and the program manipulations (e.g. hybridization, mutation) were unwieldy and fragile, with low validity rates that prohibited effective dataset scaling and diversification. This limited the breadth of MetaDB and MetaBench, while curtailing the efficacy of MetaAssist.

To address this, we designed a higher-level language that became MetaDSL-v0. This approach had a compact, modular, bilevel design that was embedded within Python and thus permitted semantically meaningful content; as such, it solved the context length and human editability issues of ProcMeta. It allowed for relative positioning, which mitigated the issues with coordinates while improving components' reusability. It also allowed for dataset augmentation through programmatic mutation, and improved the efficacy of VLM-based hybridization and mutation – we attributed this jump to our Python embedding, as VLMs show great facility with Python. Still, MetaDSL-v0 remained fragile: generated programs frequently failed, and database augmentations showed limited diversity.

Analysis of MetaDSL-v0's failure modes offered several insights; we arrived at the current MetaDSL by addressing each in turn. First, we noticed that VLMs often used hallucinated synonyms, such as `TOP_LEFT` vs `LEFT_TOP`; we added overloads for all reasonable variations of our functions and attributes. We also found that it was critical to abrogate as much spatial reasoning from the VLM as possible: a full 1/3 of failures were due to the VLM's improper positioning of vertices that form the concrete polytope tiles. We circumvented this through abstracted tile embedding functions, which generate valid embeddings from simple, meaningful parameterizations. In our final large-scale change, we swapped the relative order of lifting functions and tile embeddings (previously Embed then Lift; now, Lift then Embed). This change improved the modularity and compositionality while reducing verbosity – for example, this change allows multiple skeletons to reside in a shared Tile embedding, such that they can be patterned as a single unit. This change also paved the way for patterning of more diverse geometry-generation methods in future extensions. As a result, MetaDSL showed dramatic improvements in generation/mutation rates, and – in turn – significantly more diverse LLM-driven hybridizations.

## D    MetaDB

### D.1    Database Layout

MetaDB is structured into 4 primary directories:

- literature: Literature references that are the sources for hand-authored models.
- models: MetaDSL programs and their outputs.
- generators: Programs that create and augment models
- benchmark: The MetaBench benchmark

Data items in MetaDB can reference other items by path. These paths are either absolute (start with a forward slash "/") or relative (no leading slash). Absolute paths are assumed to start at the root of the database structure. For example, a model may reference the paper that defined it in its sources as `/literature/....`

### D.2    Provenance Information

Each Model in MetaDB starts with a triple-single-quote (`'''`) delimited yaml string called the header-block. This contains useful metadata about the program, including provenance information about how it was created, and what sources it draws on. Provenance information is recorded in two places in the header block.

The primary location is in the "sources" key. This is a dictionary where the keys are MetaDB paths to literature, models, or generators that are the source of this model. The secondary location is in `file_info→generator_info`. For models that are autogenerated via enumeration or augmentation this section contains a MetaDB path to the script that generated the file, the arguments that were passed into that script, and specific `structure_details` that specified this particular model.

### D.3 HYBRIDIZATION IMPLEMENTATION

We hybridized hand-authored models using calls to OpenAI's o4-mini model using a reasoning effort of "medium". For every pair and triplet of authored models, we used the following prompt template:

```
You have access to a DSL whose specification is as follows:
{ api_description }

I want you to help discover unique new programs. Do this by genetic crossover based on these parent Metagen DSL programs:

1)
``` python
{program 1 code}
```

2)
``` python
{program 2 code}

Combine relevant structural / logical features from each sample into one coherent DSL program.
Be sure to:
− Respect the DSL syntax strictly.
− Maintain correctness in the final structure definition.
− Keep the final program well−formed and ready to be run as a standard Metagen DSL generator.
− Provide minimal descriptive comments.

Return only the resulting code in a single code block.
```

where `api_description` is the MetaDSL API specification given in Section H, and the program code is listed excluding the header block.

### D.4 MUTATION IMPLEMENTATION

Our mutation strategy is orchestrated by a Python script that uses dynamic program analysis to identify possible swaps within a program (e.g., beam lifting procedure to shell lifting procedure) and then apply a subset of them according to user-specified probability distributions. The possible swaps are determined by comparing the preconditions for a particular target operation to the state of the seed `Structure` in question; if all preconditions are met, the swap is considered possible. For example, the seed structure shown in Figure 2(b) has a `UniformBeams` lifting function; however, based on the fact that the constituent skeleton forms a closed loop, that lifting function could be replaced with `UniformDirectShell`. However, it does *not* meet the preconditions for `UniformTPMSShellViaConjugation`, because that additionally requires that every vertex of the skeleton resides on a CP edge, which is *not* the case for our seed structure.

First, our script loads a DSL model from file and constructs the corresponding Structure object in memory. Then, it is able to modify the structure along 4 different axes. Two of the axes allow discrete adjustments: (1) switching any `Polyline` to a `Curve` or vice versa; and (2) selecting a different lifting procedure from the set of options compatible with the skeleton (as inferred by our type system). The remaining modification axes permit continuous variations: (3) repositioning a vertex within its CP element; and (4) selecting a different thickness specification for any lifting procedures. To generate a given variant, each modification axis was permitted with a pre-specified probability; we used $\Pr = 0.7$ for both discrete changes, $\Pr = 0.9$ for vertex perturbation, and $\Pr = 0.98$ for thickness perturbation. Once a given perturbation category was permitted, we looped over each opportunity for said modification within our structure specification, and evaluated a random number against the same respective probability to decide whether this specific instance should be modified or not. For example, with $\Pr = 0.7$ we allow `Polyline`/`Curve` swaps in the variant; then, each time a candidate `Polyline`/`Curve` is identified, we enact the swap with $\Pr = 0.7$. Once an instance has been approved, the specific replacement value was chosen at random from

the appropriate set of options (if more than one available). The updated structure is then written to file using the `dslTranslator`, which writes a DSL model from a Structure object. Additional mutation procedures could be implemented to further increase the vawriety of resulting structures.

Provenance Information is stored in the `sources` section of each program's header block. This is a dictionary where the keys are database paths.

### D.5   MATERIAL PROPERTIES & HOMOGENIZATION

Our simulation provides the $6 \times 6$ elastic tensor $C$ in Voigt notation, along with the compliance matrix, $S = C^{-1}$. From this, we extract 18 common material properties:

- $E$: Young's Modulus, Voigt-Reuss-Hill (VRH) average, relative to $E_{\text{base}}$.
- $E_1, E_2, E_3$: Directional Young's Moduli, relative to $E_{\text{base}}$
- $G$: Shear Modulus (VRH average), relative to $E_{\text{base}}$
- $G_{23}, G_{13}, G_{12}$: Directional Shear Moduli, relative to $E_{\text{base}}$
- $\nu$: Poisson ratio (VRH average)
- $\nu_{12}, \nu_{13}, \nu_{23}, \nu_{21}, \nu_{31}, \nu_{32}$: Directional Poisson ratios
- $K$: Bulk modulus (VRH average), relative to $E_{\text{base}}$
- $A$: Anisotropy (universal anisotropy index)
- $V$: Volume Fraction.

Our material properties are obtained through the homogenized simulation of (Makatura et al., 2023). Although imperfect (Perroni-Scharf et al., 2025), this is an accepted standard in metamaterial design – recently described as "a backbone of the recent progress in [data-driven metamaterial design]," even in spite of its inherent limitations (Lee et al., 2024). This conviction is supported by considerable physical validation of the predicted bulk behaviors (Babaee et al., 2015; Lumpe & Stankovic, 2021; Schumacher et al., 2015; Zhu et al., 2017). Of particular note, Panetta et al. (2015) – the paper which informed our/ProcMeta 's simulator – shows compelling agreement between homogenized properties and experimental measurements for Young's modulus, Poisson ratio and isotropy; these represent 3/6 core material properties in our database. The use of homogenization is furthered by the fact that there are few tractable alternatives when collecting 3D metamaterial data at scale. Surrogate models are imprecise with less explicit assumptions/limitations, while full simulations (with many repeated units) are prohibitively expensive, yet similarly prone to errors from e.g. inaccurate boundary conditions (Perroni-Scharf et al., 2025). Physical experiments are even more demanding, especially since (to preserve the full benefits) they must be individually and perhaps severally performed for every variation of interest.

Naturally, if the opportunity arises, it would be highly beneficial to incorporate additional simulated or experimental property evaluation.

### D.6   ENSURING METADB QUALITY

MetaDB is founded on a strong basis of expert programs, including 50 hand-authored examples sourced from diverse, singularly-developed designs in metamaterial literature. This large, diverse collection of seeds is unique to MetaDB, as most large datasets are derived exclusively from a small set of procedural generators. For example, Xue et al. (2025) creates a database of 180k samples, 78% of which stem from variations of the topologies in Elastic Textures (Panetta et al., 2015). The remaining 22% stem from similar generators for planar- and curved-shell structures (Liu et al., 2022; Sun et al., 2023a). Because of the reliance on such generators, Xue et al. (2025) does not offer any representation of e.g. CSG-style structures like the Bucklicrystal of Babaee et al. (2013). However, the bucklicrystal is part of our database, as shown in Figure 4(i), center). MetaDB also already includes Elastic Textures, and similar generators could be implemented for the remaining sources mentioned above.

To ensure that MetaDB only contains high-quality material definitions – even when automatically generating a large portion of our entries – material models are only added after they have passed a series of basic checks. Presently, this includes 3 criteria:

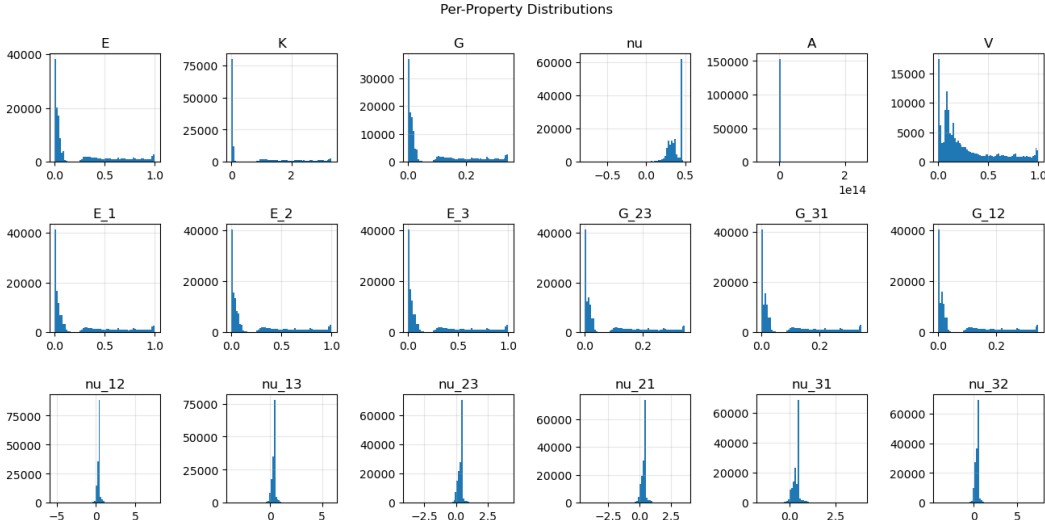

Figure 12: MetaDB material property distributions.

Table 3: Effect of sampling temperature on benchmark scores. It has no significant effect benchmark performance. Testing was done on an earlier variant of the MetaAssist (Nova) model.

| Category | Inverse Design | | Material Understanding | | Reconstruction | |
| Metric | Error ↓ | Valid ↑ | Error ↓ | CD ↓ | IoU ↑ | Valid ↑ |
| Temperature | | | | | | |
| --- | --- | --- | --- | --- | --- | --- |
| 0.00001 | .028 ± .003 | **92.7%** | **.030 ± .004** | .045 ± .001 | **.334 ± .007** | 86.7% |
| 0.7 | **.026 ± .002** | 91.4% | .032 ± .005 | **.045 ± .001** | .334 ± .007 | **87.2%** |
| Random | .028 ± .003 | 91.9% | .031 ± .004 | .045 ± .001 | .330 ± .007 | 87.2% |

- **MetaDSL compilation:** the model must contain valid python code that successfully evaluates to a MetaDSL Structure object. This includes all runtime type checking done by MetaDSL.

- **Valid Geometry Generation:** after the MetaDSL Structure object is transpiled into the target geometry kernel (in our case, ProcMeta), the kernel is run. We check the resulting geometry for validity, as measured by a non-null result that is tilable in 3D. To determine tilability, we tile the base cell in a $3 \times 3 \times 3$ lattice, then check that the boundaries are periodic and that at least one connected component of this larger base cell reaches all boundaries.

- **Physically Consistent Simulation Results:** the simulator must return reasonable results that obey physical constraints. For example, since our simulation is normalized by the base material's Young's modulus $E_{\text{base}}$, it must be the case that our simulation returns $E \leq 1$.

### D.7 METADB STATISTICS

MetaDB covers a wide range of material properties, illustrated here in as histograms (Figure 12) and as parallel coordinates (Figure 13). MetaDB has dense coverage over most of its range of elastic moduli, mid-range coverage of Poisson ratios, and dense coverage of low-anisotropy materials.

## E FURTHER BENCHMARK RESULTS

### E.1 ABLATIONS

We conducted two ablation studies.

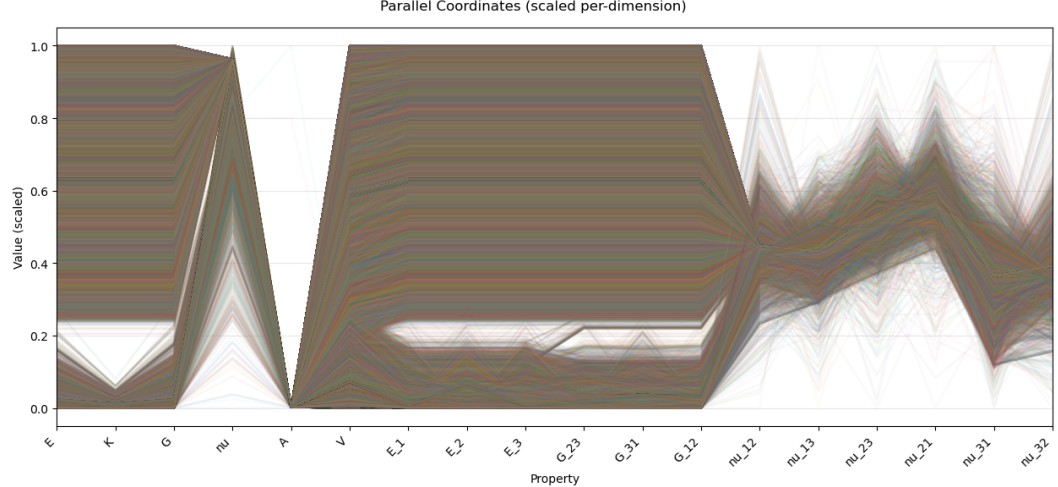

Figure 13: Properties of all MetaDB materials: each horizontal polyline is one material.

Table 4: Multi-task vs Single Task Training. **Bold** indicates significance (pair t-test, $p < .05$). Across tasks and models, multi-task training is almost always beneficial.

| Category | Inverse Design | | Material Understanding | Reconstruction | | |
| Metric | Error ↓ | Valid ↑ | Error ↓ | CD ↓ | IoU ↑ | Valid ↑ |
| Model | | | | | | |
| Omnitask MetaAssist (LLaVA) | **.022 ± .004** | 98.3% | .033 ± .006 | **.031 ± .003** | **.505 ± .029** | 94.2% |
| Single-task MetaAssist (LLaVA) | .041 ± .005 | **100%** | .035 ± .005 | .058 ± .003 | .180 ± .017 | 91.8% |
| Omnitask MetaAssist (Nova) | .018 ± .004 | **88.5%** | .017 ± .003 | **.034 ± .003** | **.463 ± .029** | 97.8% |
| Single-task MetaAssist (Nova) | .021 ± .005 | 84.2% | .024 ± .005 | .040 ± .003 | .389 ± .028 | 93.6% |

### E.1.1 VLM TEMPERATURE

To explore the effect of temperature, we use our finetuned model – MetaAssist (Nova) – to evaluate the test set with three different temperature settings: 0.00001 (as close to zero as permitted), 0.7 (Amazon's default for the Nova model) and Random (in which each sample within the test set was run with a randomly-chosen temperature). The Random category effectively allows us to test many different temperature settings, over smaller test sets each. This is done as a cost saving measure. As shown in Table 3, the sampling temperature has no significant effect on benchmark performance.

### E.1.2 MULTI-TASK VS. SINGLE TASK TRAINING

To determine whether our tasks are synergistic for finetuned models, we conduct an ablation that compares MetaAssist models trained on omnitask data versus models trained on only a single task from the dataset.

We used the MetaBench training set to produce 4 fine-tuned variants over each of the 2 base models (LLaVA and Nova). This included one OmniTask model trained over all training examples in MetaBench, and three SingleTask variants trained over one category-representative task type each (4-view reconstruction, 4-target inverse design, and multi-view-plus-code material understanding). As shown in Table 4, multi-task training is almost always beneficial, showing gains across nearly all tasks and models. Note that Table 4 condenses the SingleTask variants of each model into a single row for compactness.

### E.2 CATEGORY SUB-TASK RESULTS

Tables 5, 6, and 7 break down Table 1 for each task category into its task variations (number of views, targets, etc.). These allow a more nuanced view of MetaAssist's capabilities.

Table 5: Reconstruction Results Broken Down by task type.

| Task | 1 View | | | 2 View | | | 3 View | | | 4 View | | |
|------|--------|--------|-------|--------|--------|-------|--------|--------|-------|--------|--------|-------|
| Metric | CD↓ | IoU↑ | Valid | CD↓ | IoU↑ | Valid | CD↓ | IoU↑ | Valid | CD↓ | IoU↑ | Valid |
| Model | | | | | | | | | | | | |
| LLaVA | **0.035** | **0.456** | 93.9% | **0.032** | **0.498** | 94.2% | **0.029** | **0.519** | 94.6% | **0.031** | **0.505** | 94.2% |
| Nova | 0.037 | 0.424 | **97.7%** | 0.035 | 0.454 | **97.6%** | 0.035 | 0.464 | **97.2%** | 0.034 | 0.463 | **97.8%** |
| NovaBase | 0.119 | 0.049 | 18.7% | 0.117 | 0.050 | 17.0% | 0.118 | 0.053 | 22.0% | 0.125 | 0.050 | 25.0% |
| OpenAIO3 | 0.052 | 0.150 | 36.8% | 0.055 | 0.141 | 58.9% | 0.052 | 0.151 | 62.6% | 0.052 | 0.155 | 68.5% |

Table 6: Inverse Design Results broken down by task type.

| Task | 1 Target | | 2 Target | | 3 Target | | 4 Target | | 5 Target | | 6 Target | |
|------|----------|-------|----------|-------|----------|-------|----------|-------|----------|-------|----------|-------|
| Metric | Error↓ | Valid | Error↓ | Valid | Error↓ | Valid | Error↓ | Valid | Error↓ | Valid | Error↓ | Valid |
| Model | | | | | | | | | | | | |
| LLaVA | 0.004 | **100%** | 0.024 | **100%** | 0.021 | **99.0%** | 0.022 | **98.3%** | 0.020 | **97.3%** | 0.021 | **97.7%** |
| Nova | 0.020 | 90.0% | **0.019** | 92.8% | **0.016** | 88.0% | **0.018** | 88.5% | **0.017** | 88.5% | **0.019** | 91.1% |
| NovaBase | — | 0.0% | 0.164 | 3.0% | 0.044 | 4.2% | 0.042 | 2.1% | 0.034 | 2.8% | 0.073 | 2.3% |
| OpenAIO3 | **0.000** | 35.0% | 0.033 | 44.7% | 0.025 | 39.1% | 0.031 | 35.7% | 0.024 | 37.7% | 0.031 | 35.8% |

In reconstruction (Table 5), we see that having more viewpoints generally improves reconstruction accuracy, though this tops out for LLaVA at 4 viewpoints. We also see that multiple viewpoints greatly improves o3's validity. In practice, this is because it frequently produce 2D structures when only given a single side-view, which do not meet our periodicity requirements for validity.

For the inverse design tasks in Table 6, there is a slight trend that an intermediate number of targets is easier than very few or very many. Our hypothesis is that with a small number of targets it is possible that all targets are correlated (e.g. elastic moduli), but this is eventually counteracted by having more targets to hit. More in-depth study is required to deduce why this happens.

The expanded material understanding results shown in Table 7 reveals a difference between Nova and LLaVA; Nova is more able to take advantage of the extra image and code information, whereas LLaVA does best with only a single image. This mirrors the degredation of LLaVA in reconstruction at 4 images.

## E.3    RESULT GALLERIES

We also present randomly[2] sampled queries for each task, and visualize their results across models, along with their benchmark metrics. This shows the qualitative differences between the models' performances, while grounding the numeric metrics to make them more understandable.

Figure 14 illustrates reconstruction from 4 viewpoint renders. Of particular interest is the o3 column on the far right. For 4/5 examples, o3 correctly reproduced the basic shape of the side-on views up-to the number of repeats. This suggests that it can correctly build skeletons, but struggles with selecting the correct embedding scale.

Figure 15 illustrates material prediction based on specified property requirements. In these examples, the LLaVA models successfully generate materials that meet the given criteria, but other models occasionally generate invalid materials or fail to satisfy the specified requirements.

---

[2]rejection filtered so that all models had valid outputs for the input, except for inverse design where this was not possible

Table 7: Material Understanding results broken down by task type.

| Task | 1 View | 4 View + Code |
|------|--------|---------------|
| Metric | Error ↓ | Error ↓ |
| Model | | |
| LLaVA | 0.028 | 0.033 |
| Nova | **0.024** | **0.017** |
| NovaBase | 0.206 | 0.192 |
| OpenAIO3 | 0.083 | 0.071 |

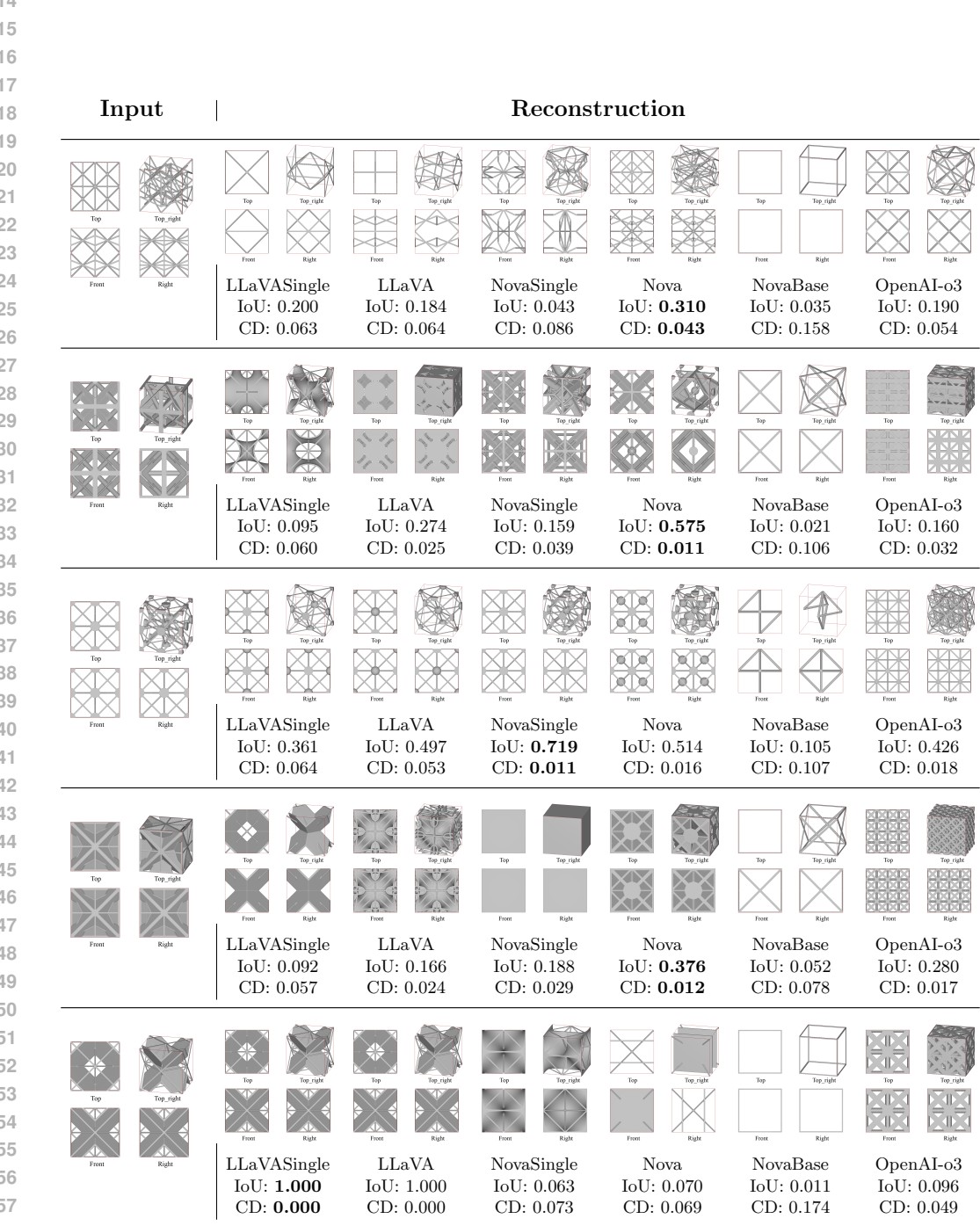

Figure 14: 4 View reconstruction results for random test samples by model. Left: the input renders shown to each model. Right: renders of predicted reconstructions.

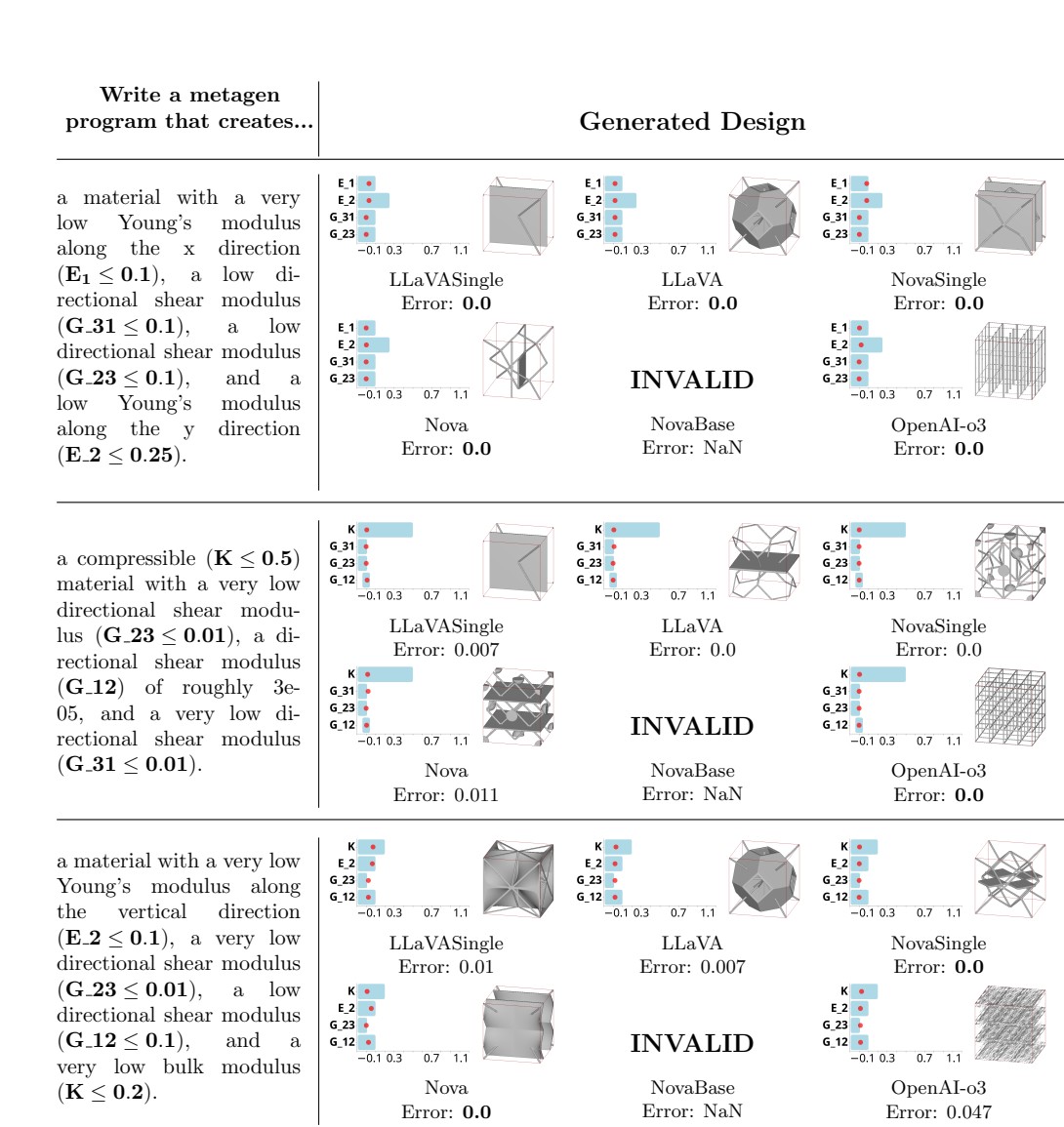

Figure 15: Inverse design results for a random selection of queries. Left: the text query given to each model. Right: paired data showing – for each model – an image of the generated structure alongside a property profile comparison. This profile shows the target values/ranges (in blue), versus simulated properties of the predicted materials (in red). Red arrows indicate that the predicted value is beyond the chart boundaries. Some models failed to produce a valid model for certain queries, indicated by the label "INVALID".

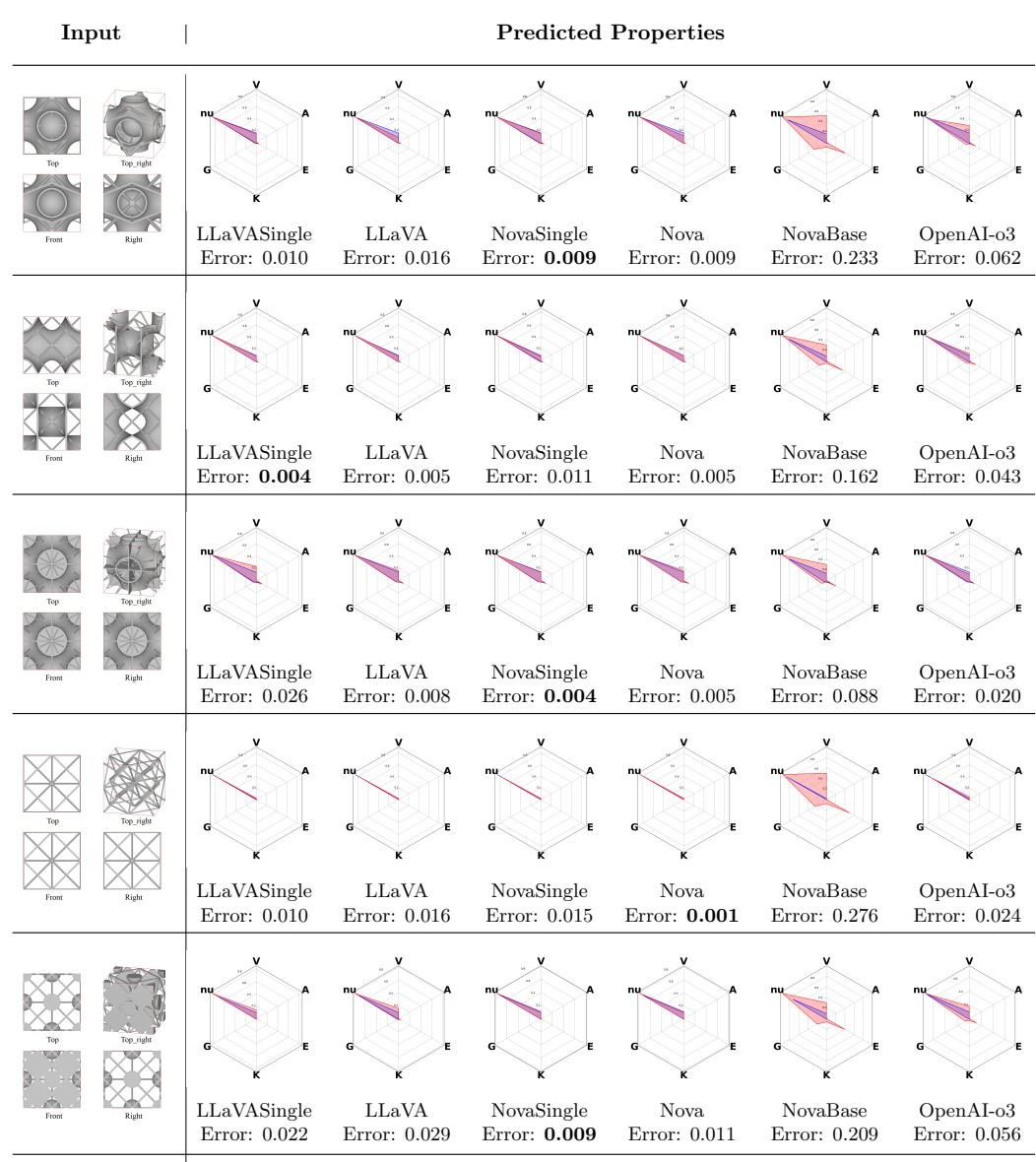

Figure 16: Material property predictions given 4 input views (shown) and the program code (not shown). The radar charts plot the 6 averaged property values (scaled and shifted to always be positive). The blue regions show the ground truth values, while red shows the prediction.

Figure 16 illustrates generated materials' predicted versus actual properties. In these examples the LLaVA and OmniTask Nova models do quite well, but single task Nova and untuned models (Novalite and o3) fall behind.

# F  METABENCH

## F.1  INTERMEDIATE REPRESENTATION

Each dataset is given by a set of .jsonl files: one file each for train, validate, and test. Each line of a .jsonl file describes a single example using a dictionary with the following keys:

- **'task_type'**: a string identifying the task category; in our case, it is one of {'reconstruction', 'inverse_design', 'material_understanding'}.

- **'label'**: unique text label identifying this task entry, using descriptive elements where applicable, such as provided image viewpoints or source files.

- **'source'**: [if applicable] path to the source metamaterial, relative to the database root (and including the leading '/')

- **'data'**: any and all data required to run evaluations, including references for large elements (e.g. images, meshes, etc.) and/or directly embedded values.

- **'query'**: natural language framing of the question to be provided to an LLM. Any images (or other non-text input) must be specified by reference.

- **'response'**: [optional] an expected response from an LLM that has been asked 'query'. This field is permitted to exist for a test example; removal of this information is the responsibility of the LLM-specific formatters, when required.

The system prompt has been purposefully excluded, both because it would be very large, and because that is an implementation detail of a predictive model, and not part of the benchmark itself.

## F.2 TASK CONSTRUCTION FOR INVERSE DESIGN

Inverse design tasks are specified as a collection of target values or bounded-ranges for a subset of material properties, from which we construct a natural-language query that describes that set of targets. Creating these tasks has two stages: selecting a set of targets, and generating an grammatically correct English sentence from those targets.

**Property References**  To aid in this process, we generate a reference dictionary with information about each of the 18 properties, of the following form:

```
{
'nu': {
    "full_prop_name": "Poisson ratio",
    "alternate_symbols": ["nu_{VRH}"],
    "property_generality": PropertyGenerality.OVERALL,
    "property_type": PropertyType.POISSON_RATIO,
    "dataset_coverage": {
        "min": -0.5,
        "max": 0.5,
        "q1": 0.3,
        "q3": 0.36,
        "densely_populated_ranges": [[0.2, 0.4]]
    },
    "smallest_meaningful_quantization": 0.01,
    "adjective_descriptors": [{"description": f"auxetic", "target_type": TargetType.UPPER_BOUND, "target_value":0}],
    "property_descriptors": [{"description": f"a negative Poisson ratio", "target_type": TargetType.UPPER_BOUND, "target_value":0},
                            {"description": f"a positive Poisson ratio", "target_type": TargetType.LOWER_BOUND, "target_value":0}],
    "verb_descriptors":      [{"description": f"contracts transversely under axial compression", "target_type": TargetType.UPPER_BOUND, "target_value":0},
                            {"description": f"expands transversely under axial compression", "target_type": TargetType.LOWER_BOUND, "target_value":0},
                            {"description": f"contracts in other directions when compressed along one axis", "target_type": TargetType.UPPER_BOUND, "target_value":0},
                            {"description": f"expands in other directions when compressed along one axis", "target_type": TargetType.LOWER_BOUND, "target_value":0},
                            {"description": f"expands transversely under axial elongation", "target_type": TargetType.UPPER_BOUND, "target_value":0},
                            {"description": f"contracts transversely under axial elongation", "target_type": TargetType.LOWER_BOUND, "target_value":0},
                            {"description": f"expands in other directions when stretched along one axis", "target_type": TargetType.UPPER_BOUND, "target_value":0},
                            {"description": f"contracts in other directions when stretched along one axis", "target_type": TargetType.LOWER_BOUND, "target_value":0}]
},
}
```

The full listing for all 18 properties is available in the metagen code provided in the supplement: `metagen/benchmarks_inverse_design.py`.

These entries provide information about the property ranges, dataset coverage, and interesting value breakpoints together with phrases that might be used to request them (e.g., "auxetic" implies $\nu < 0$). All aspects of these reference entries will be used in the following subsections to construct robust, varied and meaningful property queries for different material examples.

**Active Property Selection**  For a given structure, we enforce that the "active" property subset follows two rules. First, the active set may only employ the overall values *or* the directional values for any given property – e.g., if a profile includes measure(s) for Young's modulus, it may either include the overall Young's modulus $E$ *or* one or more of the directional values $\{E_1, E_2, E_3\}$; however, it is not permitted to simultaneously include $E$ and one or more directional variants. Moreover, a profile is only allowed to use directional variants if it is sufficiently anisiotropic. We chose our anisotropy threshold as $A \geq 0.0025$, based on a manual exploration of the correlation between material spheres and anisotropy values appearing in our dataset. Subject to these rules, we select the "active" subset of properties based on a heuristic that determines the most interesting or salient properties of a given model.

We construct this heuristic score by examining individual properties of a model, and assigning a reward or penalty based on the expected notability of a particular characteristic or combination thereof. For example, if a material is near isotropic ($A < 0.0025$), we strongly reward the anisotropy property (so it is likely to end up in the active set) and heavily penalize all directional properties (so they will not be activated, as they are not likely to be notable). If the material is sufficiently anisotropic, we look at each property with directional variants, then compute pairwise differences between the values (e.g. $E_1$ vs. $E_2$). The directional properties are rewarded proportionally to each pairwise difference, so directions with larger discrepancies are more likely to be activated. Independently, we examine the ratio between the Young's modulus $E$ and the volume fraction $V$ – if the ratio is high (i.e., the material preserves stiffness with dramatically less material / lighter weight, which is a highly sought after combination), we strongly reward both properties. Finally, we examine each property in turn, and award additional points if they exhibit values that are extreme and/or underrepresented in our dataset. The reward is proportional to the relative extremity and inversely proportional to representation.

Given these scores, we iteratively select the highest-reward properties that preserve our overall active set rules. To ensure some variation in our inverse design profiles, we also introduce the opportunity to add randomly chosen properties into our profile: after each active set addition from the ranked data, we break the loop with some low probability (10%) and fill the remaining slots with randomly chosen properties that respect the rules relative to our partial active set.

**Active Property Target Selection**  For each active property, we must now select a target value or range. To do this, we evaluate the options present in our reference dictionary, and extract all targets that are satisfied by the material at hand. We organize these into groups based on value and target type (range, value, lower/upper bound). Then, we choose the group that offers the tightest bound relative to the current material's property value. If multiple bound types are associated with the chosen target value, we select a bound type at random. Finally, we construct a profile with all targets matching the selected value and bound type. Assuming an example material where the Poisson ratio $\nu = -0.1$, the resulting profile might be as follows:

```
1  {
2      "property": "nu"
3      "target_value": 0
4      "target_type": "upper_bound"
5      "target_descriptions": [
6          {
7              "description": "auxetic",
8              "description_type": "adjective"
9          },
10         {
11             "description": "a negative Poisson ratio",
12             "description_type": "noun"
13         },
14         {
15             "description": "contracts transversely under axial compression",
16             "description_type": "verb"
17         },
18         {
19             "description": "contracts in other directions when compressed along one axis",
20             "description_type": "verb"
21         },
22         {
23             "description": "expands transversely under axial elongation",
24             "description_type": "verb"
25         },
26         {
```

```
27        " description " : "expands  in  other   directions   when stretched  along  one axis",
28        " description_type ": "verb"
29      }
30    ]
31 }
```

**Query Construction**   We want to create varied sentence structures to train and test against. To do this, each target type (value, upper bound, or lower bound) and target property has associated with it several descriptive phrases, as shown in the profile above. These phrases are paired with a part of speech (adjective, noun, or verb). As examples "very dense" (adjective), "contracts in the X direction when the Y direction is stretched" (verb), or "a negative Poisson ratio in at least one direction" (noun). Phrases that do not include numeric targets are accompanied by a parenthetical aside given a target value or range (e.g. "very dense ($V > 0.8$)."

We start by randomly selecting one phrase for each target property, binning them by part of speech, then randomizing the order within bins. Adjectives are further randomly split between *front-adjectives* that precede the noun "material" ("a very dense material") and *back-adjectives* that follow it ("a material that is very dense"). We then form a query string by applying the template:

Write a metagen program that  creates  [a/an] { front_adjectives } material { back_adjectives } {verbs} {nouns}.

The template strings are augmented with part-of-speech appropriate connectors ("that is", "with", "that", "and"), and commas, depending on the parts of number of each part of speech in each position. The pronoun (a/an) as selected based on the first letter of `{front_adjectives}` if there are any, otherwise "a" for "a material".

## G   IMPLEMENTATION DETAILS

**LLaVA**   LLaVA and LLaVASingle tune Llama3-LLaVA-Next-8b Li et al. (2024); Liu et al. (2024) using low-rank adaptation Hu et al. (2022), with with $r = 16$ and $\alpha = 32$. Models were optimized using AdamW Loshchilov & Hutter (2017) with a 1e-5 learning rate and a cosine learning rate scheduler with 0.03 warm-up ratio. Single models were trained on for 1 epoch on 4 NVIDIA B100 GPUs for 2 hours each with a batch size of 16, while the generalist LLaVA model was trained for 1 epoch on 8 B200 GPUs with a batch size of 32 over 25 hours due to its significantly larger training set, and for parity with the genaralist Nova model. During inference, the temperature was set to 0 to ensure deterministic outputs. We trained LLaVA on Amazon EC2 p6-b200.48xlarge instances. Our distributed training implementation achieved performance metrics with peak GPU utilization reaching 99% per GPU and peak memory utilization at approximately 95% per GPU across the B200s' 180GB HBM3 capacity.

**Nova**   We implemented Nova full-rank supervised fine-tuning on the Nova Lite architecture (amazon.nova-lite-v1:0:300k) using Amazon SageMaker distributed training infrastructure with ml.p5.48xlarge instances, each equipped with 8 NVIDIA H100 GPUs featuring 80GB HBM3 memory. Our training configuration employed a maximum sequence length of 32,768 tokens with a global batch size of 64, utilizing the distributed fused Adam optimizer in AdamW mode with a learning rate of $5 \times 10^{-6}$, beta parameters of (0.9, 0.999), epsilon of $1 \times 10^{-6}$, and zero weight decay. The learning rate schedule incorporated 10 warmup steps followed by decay to $1 \times 10^{-6}$, while all dropout mechanisms (hidden, attention, and feed-forward network) were disabled to perform full-rank fine-tuning across all model parameters. All models were trained for 1 epoch, where the multi-domain task required approximately 15 hours using 16 P5 instances. Our distributed training implementation achieved notable performance metrics with peak GPU utilization reaching 95% per GPU, sustained utilization of 75-95% per GPU during core training phases, and stable memory utilization at approximately 43% per GPU across the H100s' 80GB HBM3 capacity.

### G.1   TRAINING CURVES

Both the LLaVA and Nova models were trained for 1 epoch. Figure 17 shows the training curves for each model, demonstrating that the LLaVA model had converged more and more quickly than the Nova model.

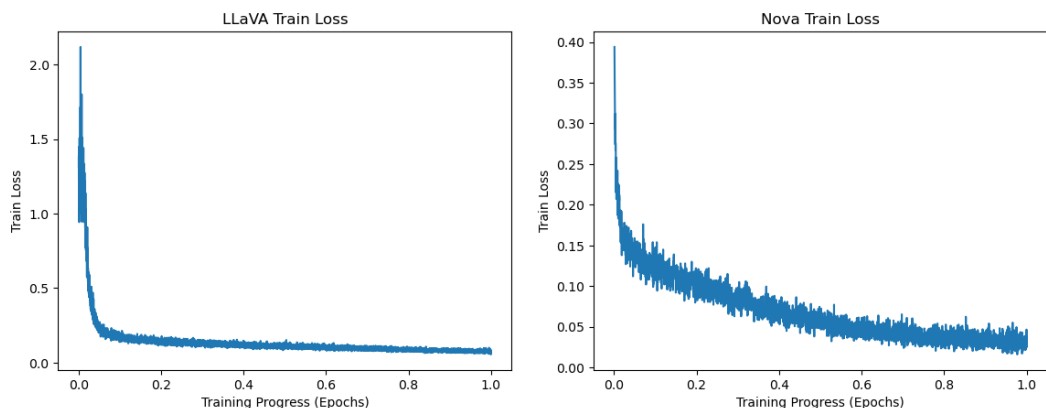

Figure 17: Training loss for LLaVA and Nova models. The smaller LLaVA model converges with significantly fewer examples than the larger Nova model.

## G.2 TIMING AND COSTS

MetaDSL execution and simulation time dominate LLM inference time for material generation. These are highly variable based on the geometric complexity of the generated program, with the majority executing and simulating in 5 minutes or less. MetaAssist generations are on average more time-complex that MetaDB (see Table 8. In practice, MetaAssist latencies are much lower because we do not run simulations in the interactive system.

| Program Source | Avg. (s) | Median (s) | Std (s) |
|---|---|---|---|
| MetaDB | 181 | 123 | 328 |
| MetaAssist | 591 | 290 | 746 |

Table 8: MetaDSL Execution and simulation times for program in MetaDB, and programs generated by MetaAssist-Nova over the MetaBench test set (reconstruction and inverse design).

Since MetaDSL is quite compact, inference can be performed efficiently with few tokens. The majority of the inference tokens are taken by the common API-description system prompt (Section H.1), the cost of which can be amortized by caching. Using NovaOmni (ignoring caching for simplicity), the average MetaBench query used 8730 tokens (8284 input and 446 output). At current API pricing, the average query would cost $0.0006, and inference for the full test set would cost $7.11.

## H QUERY TEMPLATES

For training models and running inference, we used prompt templates and inserted details for each specific query. In the following templates, $<[ \ \ldots \ ]>$ is used as a delimiter to denote the inclusion of an image.

### H.1 UNIVERSAL SYSTEM PROMPT

For consistency, every example was provided with a common system prompt that describes the Metagen DSL, explains the material properties and rendered views we have in our dataset, and describes the basic task categories.

```
You are an expert metamaterials assistant that generates and analyzes cellular metamaterial designs based on material
    properties, images, and programatic definitions in the Metagen metamaterial DSL.

# Procedural Description in a Metamaterial DSL:

{ api_description }

# Material Analysis:
```

You can analyze the density, anisotropy, and elasticity properties of metamaterials. All metamaterials are assumed to be constucted from an isotropic base material with Poisson's ratio nu = 0.45.
The Young's Modulus of this base material is not specified, instead, the elastic moduli of the metamaterials –– Young's Modulus (E), Bulk Modulus (K), and Shear Modulus (G), are expressed relative to the base material Young's modulus (E_base). This means, for example, that relative Young's Moduli can range from 0 to 1. The material properties you can analyze are:

– E: Young's Modulus, Voigt–Reuss–Hill (VRH) average, relative to E_base
– E_1,E_2,E_3: Directional Young's Moduli, relative to E_base
– G: Shear Modulus (VRH average), relative to E_base
– G_23,G_13,G_12: Directional Shear Moduli, relative to E_base
– nu: Poisson ratio (VRH average)
– nu_12, nu_13, nu_23, nu_21, nu_31, nu_32: Directional Poisson ratios
– K: Bulk modulus (VRH average), relative to E_base
– A: Anisotropy (universal anisotropy index)
– V: Volume Fraction

# Material Images:

Images of metamaterials depict a base cell of the material rendered from four viewpoints:

– from the top
– from the front side
– from the right side
– from an angle at the upper–front–right

# Tasks:

You will be asked to perform several kinds of tasks:

– Reconstruction: from one or more images of a target material, reconstruct a Metagen program that generates the metamaterial in the images.
– Inverse Design: from a description of the properties of a desired materials, write a Metagen program that creates a metamaterial with those properties.
– Material Understanding: from images of a metamaterial and/or a Metagen program, analyze a material and predict its properties.

## H.2 METADSL API

The Metagen language description (inserted as the `api_description` in the system prompt above) is as follows:

Programs in our language are built in two stages: one that creates local geometric structure, and a second that patterns this structure throughout space. Each of these is further broken down into subparts.

```
=================================
   API description  ( Boilerplate )
=================================
Each program is given as a python file (.py).
This program must import the metagen package and define a function called "make_structure()", which returns the final
       Structure object defined by the program.
If parameters are present in make_structure(), they MUST have a default value.
 Specifically, the file structure is as follows:

from metagen import *

def make_structure() -> Structure:
    <content>

=================================
   DSL description
=================================

======= Skeleton Creation ========
vertex (cpEntity, t)
    @description:
        Create a new vertex. This vertex is defined relative to its containing convex polytope (CP). It will only have an
                embedding in R3 once the CP has been embedded.
    @params:
        cpEntity      – an entity of a convex polytope (CP), referenced by the entity names.
        t             – [OPTIONAL] list of floats in range [0,1], used to interpolate to a specific position on the cpEntity.
                            If cpEntity is a corner, t is ignored.
                            If cpEntity is an edge, t must contain exactly 1 value. t is used for linear interpolation between the
                                endpoints of cpEntity.
```

```
                              If cpEntity is a face, t must contain exactly 2 values. If cpEntity is a triangular face, t is used to
                                  interpolate via barycentric coordinates. If cpEntity is a quad face, bilinear interpolation is
                                  used.

                              If the optional interpolant t is omitted for a non-corner entity, the returned point will be at the
                                  midpoint (for edge) or the centroid (for face) of the entity. Semantically, we encourage that t
                                  be excluded (1) if the structure would be invalid given a different non-midpoint t, or (2) if
                                  the structure would remain unchanged in the presence a different t (e.g., in the case of a
                                  conjugate TPMS, where only the entity selection matters).
            @returns:
                vertex          – the new vertex object
            @example_usage:
                v0 = vertex (cuboid.edges.BACK_RIGHT, [0.5])
                v1 = vertex (cuboid.edges.TOP_LEFT)

Polyline ( ordered_verts )
        @description:
            Creates a piecewise-linear path along the ordered input vertices. All vertices must be referenced to the same CP (e.g
                ., all relative to cuboid entities). The resulting path will remain a polyline in any structures that include
                it.
        @params:
            ordered_verts     – a list of vertices, in the order you'd like them to be traversed. A closed loop may be created by
                repeating the zeroth element at the end of the list. No other vertex may be repeated. Only simple paths are
                permitted.
        @returns:
            polyline          – the new polyline object
        @example_usage:
            p0 = Polyline ([v2, v3])
            p0 = Polyline ([v0, v1, v2, v3, v4, v5, v0])

Curve( ordered_verts )
        @description:
            Creates a path along the ordered input vertices. This path will be smoothed at a later stage (e.g., to a Bezier curve)
                , depending on the lifting procedures that are chosen. All input vertices must be referenced to the same CP (e.
                g., all relative to cuboid entities).
        @params:
            ordered_verts     – a list of vertices, in the order you'd like them to be traversed. A closed loop may be created by
                repeating the zeroth element at the end of the list. No other vertex may be repeated. Only simple paths are
                permitted.
        @returns:
            curve             – the new curve object
        @example_usage:
            c0 = Curve([v2, v3])
            c0 = Curve([v0, v1, v2, v3, v4, v5, v0])

skeleton ( entities )
        @description:
            Combines a set of vertices OR polylines/curves into a larger structure, over which additional information can be
                inferred. For example, within a skeleton, multiple open polylines/curves may string together to create a closed
                loop, a branched path, or a set of disconnected components.
        @params:
            entities          – a list of entities (vertices or polylines/curves) to be combined. A given skeleton must only have
                entities with the same dimension –– that is, it must consist of all points or all polylines/curves.
        @returns:
            skeleton          – the new skeleton object
        @example_usage:
            skel = skeleton ([curve0, polyline1, curve2, polyline3])
            skel = skeleton ([v0])

======= Lifting Procedures ========
UniformBeams(skel, thickness)
        @description:
            Procedure to lift the input skeleton to a 3D volumetric structure by instantiating a beam of the given thickness
                centered along each polyline/curve of the input skeleton.
        @requirements:
            The skeleton must contain only polylines and/or curves. The skeleton must not contain any standalone vertices.
        @params:
            skel              – the skeleton to lift
            thickness         – the diameter of the beams
        @returns:
            liftProc          – the lifted skeleton
        @example_usage:
            liftProcedure = UniformBeams(skel, 0.03)

SpatiallyVaryingBeams(skel, thicknessProfile)
        @description:
            Procedure to lift the input skeleton to a 3D volumetric structure by instantiating a beam of the given spatially-
                varying thickness profile centered along each polyline/curve of the input skeleton.
```

```
@requirements:
    The skeleton must contain only polylines and/or curves. The skeleton must not contain any standalone vertices.
@params:
    skel            - the skeleton to lift
    thicknessProfile - specifications for the diameter of the beams along each polyline/curve. Given as a list [ list [ floats
                ]], where the each of the n inner lists gives the information for a single sample point along the polyline/
                curve. The first element in each inner list provides a position parameter t\in[0,1] along the polyline/curve,
                and the second element specifies the thickness of the beam at position t
@returns:
    liftProc        - the lifted skeleton
@example_usage:
    liftProcedure = SpatiallyVaryingBeams(skel, 0.03)

UniformDirectShell(skel, thickness)
@description:
    Procedure to lift the input skeleton to a 3D volumetric structure by inferring a surface that conforms to the
        boundary provided by the input skeleton. The surface is given by a simple thin shell model: the resulting
        surface is incident on the provided boundary while minimizing a weighted sum of bending and stretching energies
        . The boundary is fixed, though it may be constructed with a mix of polylines and curves (which are first
        interpolated into a spline, then fixed as part of the boundary). The skeleton must contain a single closed loop
        composed of one or more polylines and/or curves. The skeleton must not contain any standalone vertices.
@requirements:

@params:
    skel            - the skeleton to lift
    thickness       - the thickness of the shell. The final offset is thickness/2 to each side of the inferred surface.
@returns:
    liftProc        - the lifted skeleton
@example_usage:
    liftProcedure = UniformDirectShell(skel, 0.1)

UniformTPMSShellViaConjugation(skel, thickness)
@description:
    Procedure to lift the input skeleton to a 3D volumetric structure by inferring a triply periodic minimal surface (
        TPMS) that conforms to the boundary constraints provided by the input skeleton. The surface is computed via the
        conjugate surface construction method.
@requirements:
    The skeleton must contain a single closed loop composed of one or more polylines and/or curves. The skeleton must not
        contain any standalone vertices.
    Each vertex in the polylines/curves must live on a CP edge.
    Adjacent vertices must have a shared face.
    The loop must touch every face of the CP at least once.
    If the CP has N faces, the loop must contain at least N vertices.
@params:
    skel            - the skeleton to lift
    thickness       - the thickness of the shell. The final offset is thickness/2 to each side of the inferred surface.
@returns:
    liftProc        - the lifted skeleton
@example_usage:
    liftProcedure = UniformTPMSShellViaConjugation(skel, 0.03)

UniformTPMSShellViaMixedMinimal(skel, thickness)
@description:
    Procedure to lift the input skeleton to a 3D volumetric structure by inferring a triply periodic minimal surface (
        TPMS) that conforms to the boundary constraints provided by the input skeleton. The surface is computed via
        mean curvature flow. All polyline boundary regions are considered fixed, but any curved regions may slide
        within their respective planes in order to reduce surface curvature during the solve.
@requirements:
    The skeleton must contain a single closed loop composed of one or more polylines and/or curves. The skeleton must not
        contain any standalone vertices.
    Each vertex in the polylines/curves must live on a CP edge.
    Adjacent vertices must have a shared face.
@params:
    skel            - the skeleton to lift
    thickness       - the thickness of the shell. The final offset is thickness/2 to each side of the inferred surface.
@returns:
    liftProc        - the lifted skeleton
@example_usage:
    liftProcedure = UniformTPMSShellViaMixedMinimal(skel, 0.03)

Spheres(skel, thickness)
@description:
    Procedure to lift the input skeleton to a 3D volumetric structure by instantiating a sphere of the given radius
        centered at vertex p, for each vertex in the skeleton.
@requirements:
    The skeleton must only contain standalone vertices; no polylines or curves can be used.
@params:
    skel            - the skeleton to lift
    thickness       - the sphere radius
@returns:
    liftProc        - the lifted skeleton
```

```
@example_usage:
    s_lift = Spheres(skel, 0.25)

======= Tile Creation ========
Tile(lifted_skeletons, embedding)
    @description:
        Procedure to embed a copy of the skeleton in R^3 using the provided embedding information. The embedding information
            can be computed by calling the "embed" method of the relevant CP.
    @requirements:
        The embedding information must correspond to the same CP against which the vertices were defined. For example, if the
            vertices are defined relative to the cuboid, you must use the cuboid.embed() method.
    @params:
        lifted_skeletons − a list of lifted skeleton entities to embed in R^3. All entities must reside in the same CP type,
            and this type must have N corners.
        embedding       − information about how to embed the CP and its relative skeletons within R^3. Obtained using the CP'
            s embed() method
    @returns:
        tile             − the new tile object
    @example_usage:
        embedding = cuboid.embed(side_len, side_len, side_len, cornerAtAABBMin=cuboid.corners.FRONT_BOTTOM_LEFT)
        s_tile = Tile([beams, shell], embedding)

======= Patterning Procedures ========
TetFullMirror()
    @description:
        Procedure which uses only mirrors to duplicate a tet−based tile such that it partitions R^3
    @params:
        N/A
    @returns:
        pat      − the patterning procedure
    @example_usage:
        pat = TetFullMirror()

TriPrismFullMirror()
    @description:
        Procedure which uses only mirrors to duplicate a triangular prism−based tile such that it partitions R^3
    @params:
        N/A
    @returns:
        pat      − the patterning procedure
    @example_usage:
        pat = TriPrismFullMirror()

CuboidFullMirror()
    @description:
        Procedure which uses only mirrors to duplicate an axis−aligned cuboid tile such that it fills a unit cube, such that
            it partitions R^3. Eligible cuboid CPs must be such that all dimensions are 1/(2^k) for some positive integer k
            .
    @params:
        N/A
    @returns:
        pat      − the patterning procedure
    @example_usage:
        pat = CuboidFullMirror()

Identity()
    @description:
        No−op patterning procedure.
    @params:
        N/A
    @returns:
        pat      − the patterning procedure
    @example_usage:
        pat = Identity()

Custom(patternOp)
    @description:
        Environment used to compose a custom patterning procedure. Currently only implemented for the Cuboid CP.
    @params:
        patternOp− outermost pattern operation in the composition
    @returns:
        pat      − the complete patterning procedure
    @example_usage:
        pat = Custom(Rotate180([cuboid.edges.BACK_RIGHT, cuboid.edges.BACK_LEFT], True,
                        Rotate180([cuboid.edges.TOP_RIGHT], True)))

Mirror(entity, doCopy, patternOp)
    @description:
```

```
      Pattern operation specifying a mirror over the provided CP entity, which must be a CP Face. Can only be used inside of
          a Custom patterning environment.
      @params:
          entity    - CP Face that serves as the mirror plane.
          doCopy    - boolean. When True, applies the operation to a copy of the input, such that the original and the
                      transformed copy persist. When False, directly transforms the input.
          patternOp- [OPTIONAL] outermost pattern operation in the sub-composition, if any
      @returns:
          pat       - the composed patterning procedure, which may be used as is (within the Custom environment), or as the input
                      for further composition
      @example_usage:
          pat = Custom(Mirror(cuboid.faces.TOP, True,
                              Mirror(cuboid.faces.LEFT, True)))

Rotate180( entities, doCopy, patternOp)
      @description:
          Pattern operation specifying a 180 degree rotation about the provided CP entity. Can only be used inside of a Custom
              patterning environment.
      @params:
          entities  - List of CP entities, which define the axis about which to rotate. If a single entity is provided, it must
                      be a CP Edge. If multiple entities, they will be used to define a new entity that spans them. For example, if
                      you provide two corners, the axis will go from one to the other. If you provide two CP Edges, the axis will
                      reach from the midpoint of one to the midpoint of the other.
          doCopy    - boolean. When True, applies the operation to a copy of the input, such that the original and the
                      transformed copy persist. When False, directly transforms the input.
          patternOp- [OPTIONAL] outermost pattern operation in the sub-composition, if any
      @returns:
          pat       - the composed patterning procedure, which may be used as is (within the Custom environment), or as the input
                      for further composition
      @example_usage:
          pat = Custom(Rotate180([cuboid.edges.FRONT_LEFT, cuboid.edges.FRONT_RIGHT], True))

Translate (fromEntity, toEntity, doCopy, patternOp)
      @description:
          Pattern operation specifying a translation that effectively moves the fromEntity to the targetEntity. Can only be used
              inside of a Custom patterning environment.
      @params:
          fromEntity- CP Entity that serves as the origin of the translation vector. Currently only implemented for a CP Face.
          toEntity  - CP Entity that serves as the target of the translation vector. Currently only implemented for a CP Face.
          doCopy    - boolean. When True, applies the operation to a copy of the input, such that the original and the
                      transformed copy persist. When False, directly transforms the input.
          patternOp- [OPTIONAL] outermost pattern operation in the sub-composition, if any
      @returns:
          pat       - the composed patterning procedure, which may be used as is (within the Custom environment), or as the input
                      for further composition
      @example_usage:
          gridPat = Custom(Translate(cuboid.faces.LEFT, cuboid.faces.RIGHT, True,
                                     Translate(cuboid.faces.FRONT, cuboid.faces.BACK, True)))

======= Structure Procedures ========
Structure (tile, pattern)
      @description:
          Combines local tile information (containing lifted skeletons) with the global patterning procedure to generate a
              complete metamaterial.
      @params:
          tile              - the tile object, which has (by construction) already been embedded in 3D space, along with all
                              lifted skeletons it contains.
          pattern           - the patterning sequence to apply to extend this tile throughout space
      @returns:
          structure         - the new structure object
      @example_usage:
          obj = Structure (tile, pat)

Union(A, B)
      @description:
          Constructive solid geometry Boolean operation that computes the union of two input structures. The output of Union(A,
              B) is identical to Union(B,A)
      @params:
          A                 - the first Structure to be unioned. This may be the output of Structure, Union, Subtract, or
                              Intersect
          B                 - the second Structure to be unioned. This may be the output of Structure, Union, Subtract, or
                              Intersect
      @returns:
          structure         - the new structure object containing union(A,B)
      @example_usage:
          final_obj = Union(schwarzP_obj, Union(sphere_obj, beam_obj))

Subtract (A, B)
      @description:
```

```
               Constructive  solid  geometry Boolean operation  that computes the  difference  (A − B) of two input  structures . The
                   relative  input order  is  critical .
           @params:
               A                    − the  first  Structure , from which B will  be subtracted . This may be the  output of  Structure , Union,
                   Subtract ,  or  Intersect
               B                    − the  second  Structure , to  be subtracted  from A. This may be the  output of  Structure , Union, Subtract ,
                   or  Intersect
           @returns:
               structure            − the  new structure  object  containing  (A − B)
           @example_usage:
               final_obj  = Subtract ( c_obj ,  s_obj )

   Intersect (A, B)
       @description :
           Constructive  solid  geometry Boolean operation  that computes the  intersection  of two input  structures , A and B.
       @params:
           A                    − the  first  Structure , which may be the  output of  Structure , Union, Subtract ,  or  Intersect
           B                    − the  second  Structure , which may be the  output of  Structure , Union, Subtract ,  or  Intersect
       @returns:
           structure            − the  new structure  object  containing the  intersection  of A and B
       @example_usage:
           final_obj  = Intersect ( c_obj ,  s_obj )

   ================================
        Prebuilt  Convex Polytopes
   ================================
   There are 3 prebuilt  convex polytopes  (CP) available  for use : cuboid, triPrism , and tet . Each CP comprises a set of  Entities ,
        namely faces , edges and corners .
   For convenience , each individual  entity  can be referenced  using the  pattern  <CP>.<entity_type>.<ENTITY_NAME>.
   For example, you can select  a  particular  edge of the cuboid with the  notation  cuboid.edges.BOTTOM_RIGHT.
   Each CP also has an embed() method which returns  all  necessary  information  to embed the CP within R^3.

   The full  list  of entities  and embed() method signatures  for our predefined  CPs are as follows :

   tet . corners .{     BOTTOM_RIGHT,
                        BOTTOM_LEFT,
                        TOP_BACK,
                        BOTTOM_BACK
            }
   tet .edges .   {     BOTTOM_FRONT,
                        TOP_LEFT,
                        BACK,
                        BOTTOM_RIGHT,
                        TOP_RIGHT,
                        BOTTOM_LEFT
            }
   tet . faces .  {     BOTTOM,
                        TOP,
                        RIGHT,
                        LEFT
                }
   tet .embed(bounding_box_side_length)
       @description :
           Constructs the  information  required  to embed the tet CP in R^3
       @params:
           bounding_box_side_length − length  of axis−aligned bounding box containing  the  tet . Float  in range  [0,1].  Must be 1/2^k
               for  some integer  k
       @returns:
           embedding        − the  embedding information .  Specifically ,  the  position  in R^3 of all  the CP corners .
       @example_usage:
           side_len  = 0.5  /   num_tiling_unit_repeats_per_dim
           embedding = tet .embed(side_len )

   triPrism . corners .{FRONT_BOTTOM_LEFT,
                    FRONT_TOP,
                    FRONT_BOTTOM_RIGHT,
                    BACK_BOTTOM_LEFT,
                    BACK_TOP,
                    BACK_BOTTOM_RIGHT
                }
   triPrism .edges.{FRONT_LEFT,
                    FRONT_RIGHT,
                    FRONT_BOTTOM,
                    BACK_LEFT,
                    BACK_RIGHT,
                    BACK_BOTTOM,
                    BOTTOM_LEFT,
```

```
                        TOP,
                        BOTTOM_RIGHT
                }
triPrism . faces .{FRONT_TRI,
                   BACK_TRI,
                   LEFT_QUAD,
                   RIGHT_QUAD,
                   BOTTOM_QUAD
                }
triPrism .embed(bounding_box_side_length)
     @description :
          Constructs the information required to embed the triangular prism CP in R^3
     @params:
          bounding_box_side_length − length of axis−aligned bounding box containing the triangular prism. Float in range [0,1].
                 Must be 1/2^k for some integer k
     @returns:
          embedding      − the embedding information. Specifically , the position in R^3 of all the CP corners.
     @example_usage:
          side_len = 0.5 / num_tiling_unit_repeats_per_dim
          embedding = triPrism .embed(side_len )

cuboid. corners .{FRONT_BOTTOM_LEFT,
                  FRONT_BOTTOM_RIGHT,
                  FRONT_TOP_LEFT,
                  FRONT_TOP_RIGHT,
                  BACK_BOTTOM_LEFT,
                  BACK_BOTTOM_RIGHT,
                  BACK_TOP_LEFT,
                  BACK_TOP_RIGHT
                }
cuboid.edges.{  FRONT_BOTTOM,
                FRONT_LEFT,
                FRONT_TOP,
                FRONT_RIGHT,
                BACK_BOTTOM,
                BACK_LEFT,
                BACK_TOP,
                BACK_RIGHT,
                BOTTOM_LEFT,
                TOP_LEFT,
                TOP_RIGHT,
                BOTTOM_RIGHT
                }
cuboid. faces .{  FRONT,
                  BACK,
                  TOP,
                  BOTTOM,
                  LEFT,
                  RIGHT
                }

cuboid.embed(width, height , depth , cornerAtAABBMin)
     @description :
          Constructs the information required to embed the cuboid CP in R^3
     @params:
          width          − length of cuboid side from left to right . float in range [0,1]. Must be 1/2^k for some integer k
          height         − length of cuboid side from top to bottom. float in range [0,1]. Must be 1/2^k for some integer k
          depth          − length of cuboid side from front to back. float in range [0,1]. Must be 1/2^k for some integer k
          cornerAtAABBMin− CP corner entity (e.g ., cuboid. corners .FRONT_BOTTOM_LEFT) that should be collocated with the cuboid'
                 s minimum position in R^3
     @returns:
          embedding      − the embedding information. Specifically , the position in R^3 of all the CP corners.
     @example_usage:
          side_len = 0.5 / num_tiling_unit_repeats_per_dim
          embedding = cuboid.embed(side_len , side_len , side_len , cornerAtAABBMin=cuboid.corners.FRONT_BOTTOM_LEFT)

cuboid.embed_via_minmax(aabb_min_pt, aabb_max_pt, cornerAtMinPt)
     @description :
          Constructs the information required to embed the cuboid CP in R^3
     @params:
          aabb_min_pt    − Minimum point of the cuboid, in R^3. Given as a list of length 3, where each component must be a
                 float in range [0,1], with 1/2^k for some integer k
          aabb_max_pt    − Maximum point of the cuboid, in R^3. Given as a list of length 3, where each component must be a
                 float in range [0,1], with 1/2^k for some integer k
          cornerAtMinPt − CP corner entity (e.g ., cuboid. corners .FRONT_BOTTOM_LEFT) that should be collocated with the cuboid'
                 s minimum position in R^3
     @returns:
          embedding      − the embedding information. Specifically , the position in R^3 of all the CP corners.
     @example_usage:
          side_len = 0.5 / num_tiling_unit_repeats_per_dim
```

```
embedding = cuboid.embed_via_minmax([0,0,0], [ side_len , side_len , side_len ], cuboid. corners .BACK_BOTTOM_RIGHT)
```

## H.3 RECONSTRUCTION

Reconstruction tasks can have any combination of one to four views. Here we only reproduced the 4 view template; the others have the irrelevant lines removed.

```
# Task:
Analyze these views of a metamaterial , then generate a metamaterial DSL procedure to reproduce it .

# Inputs :
**Rendered Views:**
Top: <[{top}]>
Front : <[{front}]>
Right : <[{right}]>
Angled (Front−Top−Right): <[{ top_right }]>

# Output Format:
Generate a Metagen program within a python code block :

``` python
from metagen import *

def make_structure (...) −> Structure :
        ...
```
```

## H.4 INVERSE DESIGN

```
# Task:
Write a metagen program that creates { query_target }.

# Output Format:
Generate a Metagen program within a python code block :

``` python
from metagen import *

def make_structure (...) −> Structure :
        ...
```
```

## H.5 MATERIAL UNDERSTANDING

Single View:

```
# Task:
Analyze these views of a metamaterial , and predict its material properties .

# Inputs :

**Rendered View:**

− Angled (Front−Top−Right): <[{ top_right }]>

# Output Format:

Output a json object , delimited by ``` json ```, where the keys are material property names, and the values are the predicted
        material properties . Predict these properties (keys):
− "A" : Anisotropy ( universal anisotropy index)
− "E" : Young's Modulus relative to E_base
− "K" : Bulk modulus relative to E_base
− "G": Shear modulus relative to E_base
− "nu": Isotropic Poisson ratio
− "V" : Relative Density (Volume Fraction)
```

Multiview + Code:

```
# Task:
Analyze these views of a metamaterial, and the Metagen program, and predict its material properties.

# Inputs:

**Metagen Program:**

{code}

**Rendered Views:**
– Top: <[{top}]>
– Front: <[{front}]>
– Right: <[{right}]>
– Angled (Front–Top–Right): <[{top_right}]>

# Output Format:

Output a json object, delimited by ''' json ''', where the keys are material property names, and the values are the predicted
        material properties. Predict these properties (keys):
– "A": Anisotropy (universal anisotropy index)
– "E": Young's Modulus relative to E_base
– "K": Bulk modulus relative to E_base
– "G": Shear modulus relative to E_base
– "nu": Isotropic Poisson ratio
– "V": Relative Density (Volume Fraction)
```

# I   EFFECT OF MODEL MERGING ON TASK PERFORMANCE

To preserve prior capabilities after target-domain tuning (e.g., safety behaviors, general reasoning, coding, tool-use), a post-hoc *merging* step is generally employed that interpolates the target fine-tuned weights with the original base model. In the main results (Table 1), we report the merged variant. However, follow-up controlled experiments indicate that an *unmerged* model can exceed the merged model on target-domain benchmarks. This appendix section formalizes the construction and quantifies the effect.

| Model | Material Understanding Error ↓ | CD ↓ | Reconstruction IoU ↑ | Valid ↑ |
|---|---|---|---|---|
| Model 1 (merge w/ base) | $0.021 \pm 0.003$ | $0.035 \pm 0.001$ | $0.449 \pm 0.007$ | $\mathbf{97.5\%} \pm 0.4\%$ |
| Model 2 (no merge) | $\mathbf{0.015} \pm 0.002$ | $\mathbf{0.028} \pm 0.001$ | $\mathbf{0.533} \pm 0.008$ | $96.4\% \pm 0.4\%$ |
| $\Delta$ (2 − 1) | $-0.006$ | $-0.007$ | $+0.084$ | $-1.1\,\mathrm{pp}$ |

Table 9: Effect of model merging on task metrics in the Nova Model. Model 1 merges the target fine-tune with the base model; Model 2 omits merging. Best values per column are in **bold**. "pp" denotes percentage points.

Let $\theta_{\text{base}}$ denote the base parameters and $\theta_{\text{tgt}}$ the target-domain fine-tuned parameters. Model 1 is constructed by weight-space interpolation with the base,

$$\theta_{\text{merge}} \;=\; (1-\lambda)\,\theta_{\text{tgt}} + \lambda\,\theta_{\text{base}} \;=\; \theta_{\text{base}} + (1-\lambda)\big(\theta_{\text{tgt}} - \theta_{\text{base}}\big), \quad \lambda = 0.6,$$

which shrinks the task-specific delta toward the base. Model 2 uses $\lambda = 0$ (no merge). All data, compute, and optimization hyperparameters are held fixed; the only difference is the merge.

Table **9** [3] shows that removing the merge yields consistently better target-domain reconstruction: *CD* improves from $0.035\pm0.001$ to $0.028\pm0.001$ (relative $\downarrow 20\%$), and *IoU* rises from $0.449\pm0.007$ to $0.533\pm0.008$ (relative $+18.7\%$). Auxiliary tasks move in the same or neutral direction: *Material Understanding Error* decreases from $0.021\pm0.003$ to $0.015\pm0.002$ (Valid $=100\%$ for both). Together, these effect sizes are large relative to the reported uncertainty and are aligned with the removal of the merge ($\lambda$: $> 0 \to 0$).

Interpolating toward $\theta_{\text{base}}$ creates a trade-off: while it can preserve broader domain capabilities, it dilutes the beneficial target-specific adaptations and reintroduces base model behaviors that perform poorly on the target task. A local quadratic approximation around $\theta_{\text{tgt}}$ with Hessian $H$ yields

---

[3]Lower is better for Error and CD; higher is better for IoU and Valid.

$\mathcal{L}(\theta_{\mathrm{merge}}) - \mathcal{L}(\theta_{\mathrm{tgt}}) \approx \frac{1}{2}\lambda^2 \|\theta_{\mathrm{tgt}} - \theta_{\mathrm{base}}\|_H^2$, predicting systematic degradation as $\lambda$ increases. The empirical ordering (Model 2 > Model 1 on CD/IoU) and the magnitude of the gains are therefore most parsimoniously explained by the *merging step* rather than by data, compute, or randomness.

Under our target focused setting, merging the target fine-tune with the base model dilutes specialization and materially harms reconstruction (CD $\downarrow 20\%$, IoU $\uparrow 18.7\%$ when omitting the merge). Consequently, we recommend $\lambda{=}0$ for target-centric deployments; alternative merge recipes may aid robustness/multitask breadth, but they are unnecessary—and harmful—for this target-domain objective.

