# OpenReview forum: "MetaGen: A DSL, Database, and Benchmark for VLM-Assisted Metamaterial Generation"
_ICLR.cc/2026/Conference — Submitted to ICLR 2026_

### Official Review · Reviewer_36hS · 2025-10-27

**Soundness:** 1
**Presentation:** 1
**Contribution:** 2
**Rating:** 2
**Confidence:** 5

**Summary:**

This work focuses on metamaterial designs, introduces a series of developments. At first, this work introduce a new hierarchical representation of metamaterial named MetaDSL, including four skeleton, tile, and pattern. After that, the authors construct a metamaterial dataset represented as MetaDSL by following previous metamaterial generation process and augment them. Finally, this work evaluates the dataset with their MetaAssist on three tasks, inverse design, material understanding, and reconstruction.

**Strengths:**

1. [Reasonable Hierarchical Data Structure] The proposed representation MetaDSL, which divides the representation of a metamaterial into skeleton, tile, and pattern, is reasonable. This data structure enables free configuration of base metamaterial structures and allows human-operable metamaterial design. In addition, the authors have clearly demonstrated the priority of MetaDSL.

2. [Data Augmentation Strategies] The proposed augmentation strategies of hybridization and mutation are well motivated by previous works, which prove the success of these data augmentation methods.

3. [Accessibility] According to the appendix, the authors implemented a usable GUI and Python package for easy use and access to the proposed dataset and DSL.

**Weaknesses:**

Although these contributions exist, this work contains many implicit issues as listed below.

Regarding Soundness

1. Unclear contribution regarding MetaAssist: This work mentions MetaAssist and regards it as one of the contributions, while there aren’t any details or definitions of it, except an unclear description, “a VLM assistant baseline and interactive CAD environment that facilitates multi-modal design interactions including language, images, geometry, and MetaDSL code.” The input, output, and training, etc. are not mentioned (including in the appendix).

2. Unrobust experiments and lack of implementation details: Especially for the benchmark part, the implementation details of all included baselines are not clear. Moreover, there is a lack of any citation or description of LLaVA, NoVA, and OpenAIO3.

3. Limited benchmarks: This paper only benchmarks and compares five baselines (where four of them are derived from MetaAssist), which is not enough for a benchmark. Notably, among the five benchmarks, only OpenAIO3 is an existing work, and LLaVA and NoVA / LLaVABase and NoVABase are derived from MetaAssist. This comparison significantly relies on their specific work; therefore, the conclusion is hard to be robust.

4. Lack of literature review for the mentioned tasks: The authors benchmark three tasks; however, they do not compare or discuss their methods with any existing works that can address these three tasks.

5. Moreover, the designed tasks are not well motivated or formally defined, except inverse design, which is generally mentioned for metamaterial design. Why reconstruction and material understanding are fundamental is not explained.

6. The material properties of this dataset are generated by homogenisation without real test and simulation, are the computed properties via homogenisation robust to reflect the real property?

Regarding Presentation

1. Challenges: The authors propose three challenges. However, except for (i), which I guess is addressed by Figure 5, they fail to demonstrate that their methods could successfully address (ii) and (iii).

2. The paper should include formal definitions of the proposed corpus, such as skeleton, tile, and pattern. In addition, they should define the whole design space of metamaterials and show why they can be represented via skeleton, tile, and pattern.

3. All notations are not well defined throughout this paper.

4. Make sure the format of subtitles is consistent, for example, paragraph titles in Section 5.1 and Section 6.2.

5. The paper should include the full name of DSL, which I guess might denote domain-specific language.

6. This MetaDSL seems based on ProcMeta; however, they are both very specific, and the authors fail to introduce and define either of them.

Regarding Significance:

Although the authors provide a large dataset with a new representation, they do not show how this dataset and representation can be used by general methods. The authors only benchmark one LLM API (OpenAIO3) and their MetaAssist, which makes it hard to demonstrate that the proposed dataset can contribute to the community.

**Questions:**

1. The position of MetaAssist in this paper. What is it and how it is used in this paper.
2. Why the authors chose the three tasks, and how they are fundamental for metamaterial design.
3. How the proposed approaches can address the proposed challenges. If possible, please provide experiments to demonstrate.

Please try to fix weaknesses

---

> ### Author Response · Authors · 2025-12-03
> **Thank you for your thoughtful comments! (1/2)**
>
> Thank you for your thoughtful and detailed review of our paper\! Your comments have helped us refine and improve our work, and we appreciate your insight. Below, we address each of your comments in turn, and **bold** all of the new experiments and/or manuscript changes. We hope that these will address your concerns, and convince you that our work meaningfully contributes to the ICLR community.
>
> ### Soundness Concerns
>
> > \[W\_S.1, Q1\] MetaAssist is unclear
>
> Thank you for this feedback\! MetaAssist is the blanket name we gave to models fine-tuned on MetaBench. As such, the input/output/training etc are precisely the descriptions provided throughout the Appendix. The interactive CAD environment is a separate, browser-based tool including a MetaDSL code-editor, a material visualization and an LLM chat environment; this is only used to interact with MetaAssist during the interactive case studies. We realized that our original descriptions and naming scheme were confusing \-- **we have updated our naming scheme and we have clarified the distinction in the paper.**
>
> > \[W\_S.2\] unclear implementation/baselines; no model citations
>
> We believe that this concern stems from the confusion caused by our original naming scheme. **We have updated the naming scheme to better match each model/experiment with their implementation details.**
>
> **We have also added the appropriate citations for LLaVA, Nova and OpenAI-o3.**
>
> > \[W\_S.3, W\_C.1\] only benchmark 4 trained models and 1 existing model (OpenAIO3)
>
> This was a miscommunication: **3 of our 5 benchmarked models are existing work,** chosen to be broadly representative of SOTA models: a large commercial VLM (Amazon Nova, originally labeled NovaBase), a large reasoning model (OpenAI-o3), and a small open-source, locally runnable model (LLaVA-Next, originally labeled LLaVABase). Only 2 of the models (originally labeled Nova and LLaVA, now MetaAssist (Nova) and MetaAssist(LLaVA)) were finetuned over MetaBench. We acknowledge that our original phrasing/naming scheme was unclear; for clarity, **we have updated the model names and descriptions in Section 6.2, and explicitly categorized each model (existing vs. specialized).** Per our results, none of the base models produce meaningful responses on MetaBench; however, fine-tuning drastically improves their performance. This suggests that (1) specialized training is necessary to perform well in this domain, and (2) benchmarking on additional existing VLMs is unlikely to produce useful baselines.
>
> > \[W\_S.4\] missing discussion of existing works for the three tasks
>
> Thank you for this comment\! **We have included a new subsection in our related works (Section 2, “Metamaterial Design”) to discuss related works for each of the three tasks**. Our work is the first to tackle accessible metamaterial design, so our formulations are slightly different from previous methods. Though the works are similar in spirit, none of them are sufficiently close to serve as comparisons for our trained MetaAssist models.
>
> > \[W\_S.5, Q2\] three tasks (inverse design, reconstruction, material understanding) not well motivated
>
> Our tasks correspond to the common stages of many iterative design frameworks, namely concept development and prototyping (reconstruction, inverse design) and evaluation (material understanding). **We have added this justification to the beginning of Section 5\. We also justify each task in the new introduction and background sections.**
>
> > [W\_S.6\] use of homogenized simulation
>
> Homogenization was recently described as “a backbone of the recent progress in \[data-driven metamaterials design\]” \[Lee et al. 2024\], even in spite of its limitations \[Perroni-Scharf et al. 2025\]. It is common to rely on metamaterial properties obtained through homogenized simulation  \[e.g. Babaee et al. 2015, Lumpe & Stankovic 2021, Schumacher et al. 2015, Zhu et al. 2017\]. Of particular note, \[Panetta et al. 2015\] \-- the paper which informed our/ProcMeta’s simulator \-- does show compelling agreement between homogenized properties and experimental measurements for Young’s modulus, Poisson ratio and isotropy; these represent 3/6 core material properties in our database. **We include more detailed information in Appendix D5.**
>
> Though it is outside the scope of this paper, the database could also be extended to incorporate physical data (or the result of alternative simulations) where available.

---

> > ### Author Response · Authors · 2025-12-03
> > **Thank you for your thoughtful comments! (2/2)**
> >
> > ### Contribution/Significance Concerns
> >
> > > \[W\_C.1\] how can the new dataset and representation be used by general methods/the community.
> >
> > Our representation is a natural, high-level interface that precisely describes metamaterial geometries; it will be released for direct community use and extension. Moreover, as shown in Figure 8, the geometry is realized through a separate (and interchangeable) backend kernels. This means that **any material described via MetaDSL can be transpiled into other representations** such as voxels, tet meshes, triangle meshes or SDFs, and then incorporated into downstream datasets or methods. Triangle- and voxel-based descriptions are already included for every structure in MetaDB. **We have added this discussion in Section 3.2.**
> >
> > ----
> >
> > ### Presentation Concerns
> >
> > > \[W\_P.1, Q3\] Addressing challenges (i-iii)
> >
> > Thank you for pointing out this disconnect\! **Our new introduction has removed this challenge-based framing,** but we explain the mapping here for clarity:
> >
> > **Challenge (i):** *navigating diverse architectures*
> >
> > MetaDSL exposes a wide range of diverse geometries behind a single coherent, extensible interface (see Fig 2-4). This includes structures from 4 of the 5 “typical” representations \[Lee et al. 2024\]. It would also be possible to capture the 5th class (voxels). **These points now appear in the discussion, limitations and future work section of the updated manuscript.** MetaDSL also facilitates reliable design navigation, **as detailed in Section 6.1.**
> >
> > **Challenge (ii):** *characterizing the intricate structure–property relationship*
> >
> > Our finetuned MetaAssist models tackle this; they are supported by MetaDB and the MetaBench tasks that query this relationship in both directions via material understanding (structure to property) and inverse design (property to structure).
> >
> > **Challenge (iii):** *collating information and assets from the highly-fragmented literature base*
> >
> > All pieces of the MetaGen ecosystem address this. MetaDSL allows us to capture diverse structures. MetaDB aggregates 150k materials derived from/inspired by literature, and places them within a coherent framework while maintaining provenance information for additional context. MetaAssist (trained on MetaBench) democratizes access to this knowledge. If researchers use, contribute to and/or develop ideas within MetaGen in the future, that will further unify the field of metamaterials.
> >
> > > \[W\_P.2,  W\_P.3, W\_P.6\] include definitions/justifications: MetaDSL/ProcMeta; skeleton, tile, pattern & design space of metamaterials
> >
> > Thank you for pointing this out\! **We have expanded the background section to include a definition of the metamaterial design space, justification of the skeleton-based construction approach, and an introduction of ProcMeta. We also re-wrote the DSL description (Section 3\) to include the appropriate definitions.**
> >
> > > \[W\_P.4\] consistent subtitle format
> >
> > **We have standardized the format of our (sub)titles and paragraph headings.**
> >
> > > \[W\_P.5\] define DSL
> >
> > **We have explicitly defined DSL in the abstract and the introduction**, by including the phrase “domain-specific language (DSL)”.
> >
> > > References
> >
> > \[Lee et al. 2024\] “Data-Driven Design for Metamaterials and Multiscale Systems: A Review.” Advanced Materials, 2024\.
> >
> > \[Panetta et al. 2015\] “Elastic Textures for Additive Fabrication.” SIGGRAPH, 2015\.
> >
> > \[Schumacher et al. 2015\] “Microstructures to control elasticity in 3D printing.” SIGGRAPH 2015\.
> >
> > \[Babaee et al. 2015\] “Three-dimensional adaptive soft phononic crystals.” Journal of Applied Physics, 2015\.
> >
> > \[Zhu et al. 2017\] “Two-Scale Topology Optimization with Microstructures.” SIGGRAPH, 2017\.
> >
> > \[Perroni-Scharf et al. 2025\] “Data-Efficient Discovery of Hyperelastic TPMS Metamaterials with
> > Extreme Energy Dissipation.” SIGGRAPH, 2025\.
> >
> > \[Lumpe & Stankovic 2021\] “Exploring the property space of periodic cellular structures based on crystal networks.” PNAS, 2021\.
> >
> > -----
> > Again, thank you for your comments! We appreciate the time and effort you’ve devoted to our paper and its betterment.

---

### Official Review · Reviewer_vMGX · 2025-10-30

**Soundness:** 3
**Presentation:** 3
**Contribution:** 2
**Rating:** 4
**Confidence:** 2

**Summary:**

The authors create a database of MetaMaterials as specialized python programs that generate the 3D structures and their properties. The programs are generated with the assistance of a LLM.  They present several benchmark tasks, including predicting the metamaterial (as a python program that generates it) from a description of its properties.

**Strengths:**

This represents a significant amount of work, a thorough understanding of metamaterials, built from prior art in the domain.  The explanation seems solid, the work extensive, the presentation is fine.   The dataset is significantly larger than the past efforts they cite.  This is indeed a novel dataset.

**Weaknesses:**

The paper is, with high likelihood, a significant advancement to the field of Metamaterial design. However, its relevance to a ML venue is questionable.

1. There is no exploration of how benchmark drives innovation in multimodal ML.
2.  ML experiments are secondary to materials science contributions.
3.  The ML content primarily relegated to the appendix, not the main paper.
4. There is limited insight into model failure modes or challenges in ML tasks.

**Questions:**

Why did you structure the paper with so much material science in the main paper and ML in the appendix?
What makes metamaterial design a compelling testbed for multimodal AI research?
Can the benchmark be generalized to other domains or tasks in ML?
What unique insights does MetaBench provide about the limitations of current VLMs?
How does MetaGen contribute to advancing ML research, particularly in multimodal AI?

---

> ### Author Response · Authors · 2025-12-03
> **Thank you for your thoughtful comments! (1/2)**
>
> Thank you for your thoughtful and detailed review of our paper\! Your comments have helped us refine and improve our work, and we appreciate your insight. Below, we address each of your comments in turn, and **bold** all of the new experiments and/or manuscript changes. We hope that these will address your concerns, and convince you that our work meaningfully contributes to the ICLR community.
>
> ------
>
> > \[W0\] a significant advancement to Metamaterial design, but questionable relevance to ML venue
>
> Thank you for this comment\! In its call for papers (https://iclr.cc/Conferences/2026/CallForPapers), ICLR 2026 specifically welcomes both “datasets and benchmarks” and “applications to physical sciences (physics, chemistry, biology, etc.).” We offers strong contributions in both areas. We are the first to offer a comprehensive representation, database, and flexible benchmark generation procedure for 3D metamaterials. We also show the feasibility of learning and exploring 3D metamaterials, even for novel tasks such as inverse design that is both mesh- and simulation-free at inference time.
>
> Moreover, there is a rich history of materials-focused papers at ICLR, including no fewer than 6 from ICLR 2025 (below). The closest work from this subset is \[Mirramezani et al. 2025\], which optimizes 2D metamaterials; although their work illustrates great potential for AI-driven metamaterial design, the 2D vs. 3D limitation was a central point of concern during the ensuing discussion, which clearly indicates both interest in and a need for efficient, effective 3D metamaterial design. We tame this design space and provide a springboard for future ML on 3D metamaterials.
>
> We also chose ICLR because our work relies so heavily on the design of a suitable, VLM-friendly representation for metamaterials. **We have emphasized the need for/advantages of our approach (Section 2\) and quantified the impact of our representation as compared to its closest predecessor (Section 6.1).**
>
> \[Wu et al. 2025\] “A Periodic Bayesian Flow for Material Generation.” ICLR 2025\.
>
> \[Das et al. 2025\] “Periodic Materials Generation using Text-Guided Joint Diffusion Model.” ICLR 2025\.
>
> \[Ding et al. 2025\] “MatExpert: Decomposing Materials Discovery By Mimicking Human Experts.” ICLR 2025\.
>
> \[Mirramezani,  et al. 2025\] “Designing Mechanical Meta-Materials by Learning Equivariant Flows.” ICLR 2025\.
>
> \[Levy et al. 2025\] “SymmCD: Symmetry-Preserving Crystal Generation with Diffusion Models.” ICLR 2025\.
>
> \[Cao et al. 2025\] “SimXRD-4M: Big Simulated X-ray Diffraction Data and Crystal Symmetry Classification Benchmark.” ICLR 2025
>
> > \[W1, Q2, Q3, Q5\] Benchmark ability to advance multimodal ML, serve as test bed, generalize to other domains
>
> 3D metamaterials represent a fundamentally unsolved, multimodal problem at the frontier of materials science and engineering design. Thus, effective AI/ML solutions would be meaningful and exciting \-- similar to AI advancements in protein folding, or drug discovery. We believe that multimodal AI is the right tool for this job, and our paper provides the infrastructure and initial baselines to unlock this direction.
>
> Metamaterial design also ties into several cutting-edge lines of inquiry in AI/ML, such as:
>
> - Equivariant/Symmetry aware data processing (also useful for chemistry, engineering)
> - Reinforcement learning (RL) \-- MetaBench tasks have verifiable outputs with many levels of complexity; this provides an ample testbed for RL, curriculum learning, etc.
> - Embedding spaces and similarity metrics \-- as detailed in Appendix C.3, a given metamaterial can be represented by several different unit cells and/or underlying construction methods. There also exist many metamaterial architectures realizing a given property profile. This presents a challenging many-to-one problem for latent space design, along with interesting questions about appropriate similarity/diversity metrics. MetaDB could provide the data to study this.
> - Multi-turn agentic frameworks \-- metamaterial design is iterative in nature. Our MetaAssist models already demonstrate some support for this, so they could be used to collect and verify samples for multi-turn editing.
>
> MetaGen could provide interesting data on which to explore and advance these strategies. **We briefly mention these opportunities at the beginning of Section 7\.**

---

> > ### Author Response · Authors · 2025-12-03
> > **Thank you for your thoughtful comments! (2/2)**
> >
> > > \[W2, W3, Q1\] ML secondary to materials science
> >
> > We prioritized the materials science content for two reasons: (1) we assumed that it would be less familiar to the ML community, and thus require more description. (2) The ML techniques used during our experiments were (intentionally) limited to standard procedures, to serve as an accurate baseline of VLM capabilities. Our works main contributions lie in creating a unified, reconfigurable ecosystem ( DSL, database and benchmark generation procedures) that was suitable to kickstart VLM research on metamaterials.
> >
> > However, we did uncover several generalizable insights for VLMs, and we agree that they should be featured more prominently. **We have brought several experiments and generalizable insights back into the main paper (primarily in Section 6). We have also restructured Sections 1,2, and 4 to more heavily feature the ML aspects/insights of our work.**
> >
> > > \[W4\]There is limited insight into model failure modes or challenges in ML tasks.
> >
> > We observed sporadic evidence of model failure modes \-- for example, some of our earlier models seemed to gravitate toward certain metamaterial structures (like a Schwarz P TPMS). We also occasionally observed an overfitting-like effect, in which query formats too dissimilar from our training format seemed to generate poor performance, mirroring that of the zero-shot models. However, we did not investigate these modes deeply enough to include them in the paper.
> >
> > > \[Q4\] insights about current VLM limitations?
> >
> > Our current experiments using MetaBench reveal several insights (**now included in the Results section**):
> >
> > - Finetuning is required to support a custom, technical DSL: all of our off-the-shelf VLMs perform poorly across all tasks and metrics. This includes a small open source model (LLaVA), a large commercial model (Nova) and a reasoning model (OpenAI o3)
> > - Even after simple finetuning, finetuned models perform well. This is true for small and large models. However, the large model is significantly better at generating novel materials, while the small model often regurgitates training data.
> > - Our strong performance on the inverse design task expands the scope of VLM-assisted reverse engineering, as it demonstrates VLMs’ ability to reason over not one but two complex domain mappings (code $\\leftrightarrow$ geometry $\\leftrightarrow$ performance) across three separate modalities.
> > - A single-task vs. omni-task ablation indicates that training over several synergistic tasks improves model performance
> > - Our success frames custom DSLs as an effective medium for complex VLM-assisted design tasks.
> >
> >
> > --------
> > Again, thank you for your comments! We appreciate the time and effort you’ve devoted to our paper and its betterment.

---

### Official Review · Reviewer_QFCq · 2025-10-31

**Soundness:** 2
**Presentation:** 3
**Contribution:** 3
**Rating:** 4
**Confidence:** 3

**Summary:**

The paper presents MetaGen, a unified ecosystem for AI-assisted metamaterial design composed of four parts: MetaDSL, a typed domain-specific language for representing metamaterial structures; MetaDB, a 150k-entry database linking DSL code, geometry, renderings, and simulated elastic properties; MetaBench, a benchmark covering reconstruction, material-understanding, and inverse-design tasks; and MetaAssist, vision–language model baselines trained on MetaBench. Empirically, the fine-tuned MetaAssist variants substantially outperform base models and a zero-shot model on validity and error metrics, and analysis suggests MetaDB expands the property-space coverage beyond hand-authored designs.

**Strengths:**

This paper addresses an important and timely problem—how to enable AI-assisted metamaterial design through structured, interpretable, and verifiable representations. Building a unified ecosystem that connects language, geometry, and physical properties is both necessary and impactful for advancing the field. The proposed MetaDSL is particularly practical and well-designed, offering a compact, human- and machine-readable language that effectively bridges symbolic and physical reasoning. The paper also provides comprehensive experiments and detailed analyses, including dataset statistics, benchmark splits, and qualitative case studies, which together make the results convincing.

**Weaknesses:**

1. In Table 1, the fine-tuned MetaAssist models show substantial gains over their base models and the zero-shot OpenAI o3 baseline. However, it remains unclear whether any other models fine-tuned specifically on metamaterial data exist, and how they would perform on the same MetaBench tasks. Including such baselines would help better understand the reported improvements.

2. Although MetaDSL is presented as a general, extensible representation, the appendix states that “all of our structures have a translational unit residing in a unit cube.” I am not an expert in the metamaterial domain, but does this constraint mean that certain classes of metamaterials (like non-periodic) cannot yet be represented within MetaDSL?

3. For the inverse design tasks, the evaluation focuses on error and validity. It would be valuable to include novelty metrics to assess whether generated designs are genuinely new or simply replicate known patterns.

4. Given that the main contributions are the DSL, database, and benchmark, the impact of this work will depend heavily on publicly releasing MetaDB and the associated tools, along with clear licensing and contribution policies.

5. Minor: Some figures (e.g., Figure 5) have text that is difficult to read due to small font sizes.

**Questions:**

Please refer to weaknesses.

---

> ### Author Response · Authors · 2025-12-03
> **Thank you for your thoughtful comments! (1/1)**
>
> Thank you for your thoughtful and detailed review of our paper\! Your comments have helped us refine and improve our work, and we appreciate your insight. Below, we address each of your comments in turn, and **bold** all of the new experiments and/or manuscript changes. We hope that these will address your concerns, and convince you that our work meaningfully contributes to the ICLR community.
>
> > \[W1\] other models fine-tuned on metamaterial data
>
> To our knowledge, we are the first to use VLMs for general metamaterial design, and the first to begin from high-level specifications. VLMs have been used in a few metamaterial works, but **none are sufficiently similar to our work to serve as baselines**. For example, \[Naghavi Khanghah et al. 2024\] uses a small LLM to predict edges in a graph representation; \[Tian et al. 2025\] uses a VLM to describe 2D structures, and \[Lu et al. 2024\] tunes a model to predict electromagnetic spectra of 2.5D metasurfaces. None are trained on general 3D mechanical metamaterials. **We have extended our background section, including a new paragraph (“Metamaterial Design”) to discuss these works in relation to our own.**
>
> > \[W2\] appendix says “all of our structures have a translational unit residing in a unit cube.”; does that mean non-periodic cannot yet be represented within MetaDSL?
>
> The cited statement refers to the core MetaDSL implementation (and all materials in MetaDB): since we initially targeted ProcMeta as our backend, our materials were biased by their limitations (ie, the translational unit cube). This is not a fundamental limitation of MetaDSL: the alternate backend (Section 3.2) already allows non-unit cubes, as in Figure 3b. **We have clarified the statement in the appendix, and included a brief discussion in the main paper at the end of Section 3.2.**
>
> Aperiodic structures are currently unsupported in MetaDSL. However, these (and several others) are unimplement*ed,* not unimplement*able* \-- the semantics of MetaDSL are compatible with these materials, so future iterations may represent these classes of materials without invalidating the ones we already have.
>
> > \[W3\] novelty metrics for inverse design
>
> Thank you for this suggestion\! **We have added a novelty evaluation for the inverse design tasks in Table 1\.** For the tuned Nova model, 76% of inverse design generations are unique (do not appear in the training set, as measured by voxel equivalence). For the tuned LLaVA model, 49% are unique.
>
> > \[W4\] impact dependent on data release and licensing
>
> We agree\! We are currently working with AWS Open Data, and expect to publish all data and code under an MIT ([https://opensource.org/license/MIT](https://opensource.org/license/MIT)), Apache ([https://www.apache.org/licenses/LICENSE-2.0](https://www.apache.org/licenses/LICENSE-2.0)), or similar license. Contribution guidelines will be determined after release, to ensure that the policies are sufficiently flexible to accommodate the community needs and preferences regarding use cases, extensions, etc. **We have added a reproducibility statement to this effect.**
>
> > \[W5\] Minor: Some figures (e.g., Figure 5\) have text that is difficult to read due to small font sizes.
>
> Thank you, **we have improved the original Figure 5**, which is now a simplified inset in Section 6\.
>
> > References
>
> \[Naghavi Khanghah et al. 2024\] “Reconstruction and Generation of Porous Metamaterial Units Via Variational Graph Autoencoder and Large Language Model.” Journal of Computing and Information Science in Engineering, 2024\.
>
> \[Tian et al. 2025\] “A Multi-Agent Framework Integrating Large Language Models and Generative AI for Accelerated Metamaterial Design.” arXiv, 2025\.
>
> \[Lu et al. 2024\] “Can Large Language Models Learn the Physics of Metamaterials? An Empirical Study with ChatGPT." arXiV, 2024\.
>
> ------
>
> Again, thank you for your comments! We appreciate the time and effort you’ve devoted to our paper and its betterment.

---

### Official Review · Reviewer_qTbH · 2025-11-01

**Soundness:** 2
**Presentation:** 1
**Contribution:** 2
**Rating:** 2
**Confidence:** 4

**Summary:**

This paper introduces MetaGen, a unified framework for AI-assisted metamaterial design. It includes MetaDSL, a compact language for describing structures; MetaDB, a dataset of over 150,000 metamaterials with code, geometry, and properties; MetaBench, a benchmark for reconstruction, property prediction, and inverse design; and MetaAssist, an interactive assistant based on large vision–language models. Together, they establish a platform for studying and advancing multimodal material design.

**Strengths:**

1. This paper propose MetaDSL, a domain specific language for metamaterial. This can be regarded as a novel and concise way to express metamaterial structures.
2. This paper builds a large scale metamaterial dataset which contains 3D structures and mechanical properties. The dataset is benificial to the metamaterial community.

**Weaknesses:**

1. The motivation is not clear. The authors have not clearly explain why such a DSL is needed. Considering that they want to leverage the power of LLMs, existing textual description of metamaterial can also serve (e.g., depicting truss-based metamaterial with node positions and edges in text format). As for representation, the geometrical representation can already satisfy the need of metamaterial research, like voxel-representation.
2. As the author mentioned, there is a hurdle like "navigating the immense geometric diversity of candidate architectures". I understand that every valid "MetaDSL" code can be readily transferred into real 3D shape. However, it's not trivial that existing structures can be transferred into MetaDSL format (e.g., those in Yang et al's work[r1]). If existing structures cannot be easily transferred into MetaDSL, it's still a doubt whether MetaDSL can allow to "realize the full potential" of metamaterial.
3. It is questionable whether MetaDSL is a self-contained representation instead of the accumulation of engineering efforts. For example, the authors use "SpatiallyVaryingBeams" to accommodate truss-based metamaterial with non-uniform truss. I understand that MetaDSL can integrate further engineering techniques like this to be extended to more types of metamaterial, but to me it's not a natural and universal representation like voxels, but instead good enginnering efforts.
4. The benckmark is too specifically curated, hence limited impact. The benchmark specifies three tasks, each of them related to MetaDSL. For example, inverse design is defined as "given a target property profile, the desired output is a DSL program". The highly curated benchmark is the reason, as I recon, why only four methods are compared in this paper. It is still unclear how well other inverse design methods which start from properties and output 3D structures can perform compared to the four baselines. If such inverse design methods work better, it is questionable why MetaDSL is needed.
5. The authors claim to use "MetaAssist, a VLM assistant baseline and interactive CAD environment", but I haven't found enough description about this, especially the "interactive CAD environment".

[r1] Yang, Yanyan, et al. "Guided diffusion for fast inverse design of density-based mechanical metamaterials." arXiv preprint arXiv:2401.13570 (2024).

**Questions:**

1. Although the authors provide some statistical information regarding the properties in the dataset, I still don't have a sound understanding of the difficulty of the property prediction task. Could the authors also report the mean and standard deviation of each property, or other empirical facts like how a random guess/constant guess perform compared with the four baselines?

---

> ### Author Response · Authors · 2025-12-03
> **Thank you for your thoughtful review! (1/2)**
>
> Thank you for your thoughtful and detailed review of our paper\! Your comments have helped us refine and improve our work, and we appreciate your insight. Below, we address each of your comments in turn, and **bold** all of the new experiments and/or manuscript changes. We hope that these will address your concerns, and convince you that our work meaningfully contributes to the ICLR community.
>
> > \[W1\] Motivating MetaDSL \-- why not node/edge or voxels, which are geometrically sufficient?
>
> MetaDSL is needed because no existing representation satisfies the criteria for VLM-assisted metamaterial design \-- for example, node/edge descriptions are compact and editable, but they are limited to truss-based structures; by contrast, voxels are expressive but lacking geometric fidelity, semantic clarity and editability. MetaDSL is compact, precise and interpretable across a wide range of metamaterials (with the ability to encompass more), so it satisfies all criteria. **We have extended and reframed our Background section to elaborate on the shortcomings of existing representations, and motivate the need for/benefits of MetaDSL. We also discuss the benefits of MetaDSL in our modified Section 3, and quantify them in Section 6.2.**
>
> **Voxels:** respectfully, **we disagree with the claim that voxels are sufficient**. **Our updated background section (“Metamaterial Representation”) includes a dedicated paragraph to explain our reasoning.** We highlight that voxel-based datasets frequently stem from other complementary representations, including \[Yang et al. 2024\] (suggested by the reviewer), which uses trigonometric functions. This underscores the utility and importance of non-voxel shape representations. The degraded fidelity and mechanical performance are also dramatic: we compared a precise Schwarz P TPMS from MetaDSL with its 128^3 voxelized counterpart. The latter had a maximum stress value that was 1.77x ours; thus, even though the voxel version had twice as much material, it would encounter critical stress values under half as much load.
>
> > \[W2\] incorporating existing structures
>
> Thank you for raising this point\! We agree: our ecosystem should accommodate as many existing structures as possible. We already offer considerable diversity (Fig 2-4), and address structures given by 4 of the 5 “typical” representations \[Lee et al. 2024\]; voxels are the only unsupported class.
>
> Using our CartesianVolume extension (now, Section 3.1), it would be **conceptually trivial to incorporate a naive voxel approach into MetaDSL**. However, we intentionally omitted direct voxel specification, because it is incompatible with MetaDSL’s goal (it’s not compact, and lacks semantics). Moreover, as illustrated in \[Yang et al. 2024\], voxel metamaterials are typically well-described by (or indeed, seeded from) higher-level structures. As such, we believe that the **best way to incorporate voxelized structures is to develop converters that fit them with higher-level templates** as in \[Chen et al. 2018\]. The development of such converters is outside the scope of this paper, but we firmly believe that they are possible. **We added this discussion to our Section 7\.**
>
> > \[W3\] MetaDSL as a self-contained/natural representation vs. accumulated engineering effort.
>
> **There is no single “natural” representation** that captures all relevant designs while preserving important considerations like compactness, precision and editability; each approach has inherent tradeoffs, necessitating many different representations \[Lee et al. 2024\]. Although this plurality fragments the field, limits exploration, and prohibits data reuse \[Makatura et al. 2023, Xue et al. 2025, Lee et al. 2024, Surjadi & Portela 2025\], it persists because **each representation is uniquely well-suited to a particular subset of structures, users, or algorithms**. MetaDSL harnesses their strengths through a unified interface. In that respect, “good engineering efforts” are a key component of MetaDSL, and the only way to achieve our stated goals. **We now comment on this in the background section.**

---

> > ### Author Response · Authors · 2025-12-03
> > **Thank you for your thoughtful review! (2/2)**
> >
> > > \[W4\] Benchmark too dependent on MetaDSL; do other inverse design methods mean ours isn’t necessary?
> >
> > We built MetaBench around MetaDSL because we believe that it is the right tool for VLM-assisted metamaterial design. However, this is not strictly necessary: the task content would still be relevant and sensible if MetaDSL were replaced by e.g. voxels or a node/edge text description. We have already taken some steps to support this, **as described in our new Section 5\.**
> >
> > We agree that dedicated design methods may produce more accurate results than our approach, given a suitably transferable task; we do not intend or claim to displace them. However, our tasks are currently motivated by the goal of *democratized* metamaterial design, which allows users to start from higher-level queries while still receiving reasonable responses. Because of this, our queries do not take on a standard format that is directly suitable for known methods. **We discuss this in Section 6.2.**
> >
> > > \[W5\] MetaAssist definition unclear
> >
> > Thank you for this feedback\! MetaAssist is the blanket name we gave to models fine-tuned on our MetaBench test set. The interactive CAD environment is a browser-based tool that combines a metagen code-editor with a material visualization and an LLM chat environment to interact with MetaAssist. **We have updated our naming scheme and clarified this distinction in the paper.**
> >
> > > \[Q1\] Extra baselines for material understanding
> >
> > Thank you for this suggestion\! We compared a constant prediction of the dataset mean, random normally distributed predictions, and random uniformly distributed predictions, and found them to have .072 \+/- .007, .122 \+/- .007, and .414 \+/- .010 material understand error respectively, as compared to .021 \+/- .003 from our best MetaAssist model. Our predictions show nearly 4x improvement over the dataset mean, which is the best of the three suggested baselines by a considerable margin. **We have added these baselines to summary Table 1\.**
> >
> >
> > Again, thank you for your comments! We appreciate the time and effort you’ve devoted to our paper and its betterment.
> >
> >
> > > References:
> >
> > \[Panetta et al. 2015\] “Elastic Textures for Additive Fabrication.” SIGGRAPH, 2015\.
> >
> > \[Abu-Mualla & Huang 2025\] “Symmetry-Induced Mechanical Metamaterials Design Framework and Dataset Generation.” ASME Journal of Mechanical Design, 2025\.
> >
> > \[Makatura et al. 2024a\] “How Can Large Language Models Help Humans in Design and Manufacturing? Part 1: Elements of the LLM-Enabled Computational Design and Manufacturing Pipeline.” HDSR, 2024\.
> >
> > \[Makatura et al. 2024b\] “How Can Large Language Models Help Humans in Design And Manufacturing? Part 2: Synthesizing an End-to-End LLM-Enabled Design and Manufacturing Workflow.” HDSR, 2024\.
> >
> > \[Chen et al. 2018\] “Computational discovery of extremal microstructure families.” Science Advances, 2018\.
> >
> > \[Xue et al. 2025\] “MIND: Microstructure INverse Design with Generative Hybrid Neural Representation.” SIGGRAPH, 2025\.
> >
> > \[Lee et al. 2024\] “Data-Driven Design for Metamaterials and Multiscale Systems: A Review.” Advanced Materials, 2024\.
> >
> > \[Yang et al. 2024\] "Guided diffusion for fast inverse design of density-based mechanical metamaterials." arXiv preprint arXiv:2401.13570 (2024).
> >
> > \[Makatura et al. 2023\] “Procedural Metamaterials: A Unified Procedural Graph for Metamaterial Design.” ACM TOG, 2023\.
> >
> > \[Surjadi & Portela 2025\] “Enabling three-dimensional architected materials across length scales and timescales.” Nature Materials, 2025\.

---

### Author Response · Authors · 2025-12-03
**Summary for Area Chair (1/2)**

We thank all reviewers for their thoughtful and constructive feedback. We have made substantial revisions to address every concern raised, resulting in a significantly strengthened manuscript. Our work provides the first comprehensive ecosystem for VLM-assisted 3D metamaterial design, offering essential infrastructure—MetaDSL (representation), MetaDB (150k structure-property pairs), and MetaBench (flexible benchmark)—alongside strong baselines and actionable ML insights.

## **Core Contributions**

Our paper introduces three interconnected components that together enable VLM-based metamaterial design: (1) **MetaDSL**, a compact, interpretable domain-specific language that unifies diverse metamaterial representations while maintaining geometric precision and editability; (2) **MetaDB**, one of the largest and most diverse metamaterial datasets with paired structure-property information for 150,000 entries; and (3) **MetaBench**, a flexible benchmark generation procedure for three fundamental tasks. Critically, our approach enables inverse design (property → structure) that is both mesh- and simulation-free at inference time—a first for this domain that decimates iteration time and complexity. We demonstrate feasibility with strong baselines using both large commercial models and small open-source alternatives.

## **Major Concerns Addressed**

### **Representation Choice & Practical Utility (qTbH-W1, qTbH-W2, 36hS-W\_C.1)**

Reviewers questioned why we chose MetaDSL over existing representations like voxels or node/edge graphs. We have substantially expanded Section 2 to articulate this motivation: existing representations fail to simultaneously satisfy the criteria for VLM-assisted design (compactness, precision, editability, and expressiveness). We now include quantitative evidence: a voxelized (128³) version of a Schwarz P structure had 1.77× higher maximum stress than our precise MetaDSL version, meaning it would fail under half the load despite containing twice as much material (qTbH-W1).

Regarding practical utility, **MetaDSL can transpile to other formats** (voxels, triangle meshes, tet meshes, SDFs) for downstream use, and we already provide triangle- and voxel-based descriptions for every structure in MetaDB. This interoperability means any material described via MetaDSL can be incorporated into existing datasets and methods (36hS-W\_C.1, now Section 3.2). Our representation currently encompasses structures from 4 of 5 "typical" metamaterial classes identified in recent surveys \[Lee et al. 2024\], with pathways to incorporate the fifth (voxels) via fitting procedures as discussed in qTbH-W2.

### **ML Relevance, Insights, and Venue Appropriateness (vMGX-W0, vMGX-W2/W3, 36hS-W\_S.4)**

Reviewer 3 raised important questions about ML relevance and ICLR suitability. We emphasize that **ICLR 2026 explicitly welcomes "datasets and benchmarks" and "applications to physical sciences"** in its call for papers. Materials-focused papers have a strong presence at ICLR—we cite 6 from ICLR 2025 alone, including \[Mirramezani et al. 2025\] on 2D metamaterial optimization, where discussion highlighted the need for effective 3D approaches. Our work directly addresses this gap.

We have restructured Sections 1, 2, and 6 to prominently feature ML contributions and insights, which include:

* **Off-the-shelf VLMs fail entirely** on these tasks; specialized training is essential. This finding holds across diverse models: large commercial (Amazon Nova), large reasoning (OpenAI-o3), and small open-source (LLaVA-Next).
* **Both large and small models succeed after finetuning**, but with different tradeoffs: large models generate more novel structures (76% unique for Nova vs 49% for LLaVA), while small models show comparable accuracy on other metrics.
* **Multi-task training consistently improves performance** over single-task alternatives across all metrics.
* **Domain-specific languages are effective mediums** for complex VLM tasks requiring multiple domain mappings (code ↔ geometry ↔ performance).

We also added discussion (Section 7\) of connections to cutting-edge ML research: equivariant/symmetry-aware processing, reinforcement learning testbeds, embedding space design for many-to-one mappings, and multi-turn agentic frameworks (vMGX-Q2, Q3). These connections demonstrate how MetaGen can advance multimodal ML research.

---

> ### Author Response · Authors · 2025-12-03
> **Summary for Area Chair (2/2)**
>
> ### **Clarity, Naming, and Presentation (qTbH-W5, 36hS-W\_S.1/2, 36hS-W\_P.1-6)**
>
> Multiple reviewers noted confusion about terminology, particularly "MetaAssist." We have **completely revised our naming scheme and definitions**:
>
> * **MetaAssist(Nova)** and **MetaAssist(LLaVA)** now clearly denote our two finetuned models
> * The **interactive CAD environment** is explicitly described as a separate browser-based tool
> * All baselines are categorized: 3 existing models (Amazon Nova, OpenAI-o3, LLaVA-Next) \+ 2 finetuned variants
>
> We have also added formal definitions throughout Section 3, including explicit explanations of MetaDSL components (skeleton, tile, pattern) and an expanded background section introducing ProcMeta and the metamaterial design space (36hS-W\_P.2, W\_P.6). Citations for all models are now included (36hS-W\_S.2), and we have standardized formatting across all sections (36hS-W\_P.4).
>
> ### **Benchmark Design, Evaluation, and Task Motivation (qTbH-W4, QFCq-W3, 36hS-W\_S.3/4/5)**
>
> Reviewers requested additional baselines and clearer task motivation. We have added:
>
> * **Simple baselines** (mean, random uniform, random normal) to Table 1, showing our best model achieves nearly 4× improvement over the strongest simple baseline (qTbH-Q1)
> * **Novelty metrics** for inverse design: 76% of generations from MetaAssist(Nova) are unique (not in training set), versus 49% for MetaAssist(LLaVA) (QFCq-W3)
> * **Explicit task motivation** (Section 5): these tasks correspond to common stages in iterative design—concept development/prototyping (reconstruction, inverse design) and evaluation (material understanding) (36hS-W\_S.5, Q2)
>
> We clarify that **MetaBench is not strictly dependent on MetaDSL**: task content would remain relevant if MetaDSL were replaced by other representations like voxels or text descriptions (qTbH-W4). We have also added a new "Metamaterial Design" subsection to Section 2, comparing our approach to related work \[Naghavi Khanghah et al. 2024, Tian et al. 2025, Lu et al. 2024\] and explaining why none are sufficiently similar to serve as direct baselines (36hS-W\_S.4).
>
> ## **Additional Improvements**
>
> * **Scope clarification**: Aperiodic structures are currently unsupported but semantically compatible with MetaDSL; future iterations can add them without invalidating existing materials (QFCq-W2, now Section 3.2)
> * **Data release**: Working with AWS Open Data for MIT or Apache license; contribution guidelines will accommodate community needs (QFCq-W4, new reproducibility statement)
> * **Challenge-solution mapping**: Explicitly connected each ecosystem component to the three stated challenges (geometric diversity, structure-property relationships, literature fragmentation) with experimental evidence (36hS-W\_P.1, Q3, now Section 6\)
> * **Homogenization validity**: Addressed with citations showing compelling agreement between homogenized simulations and experimental measurements for 3/6 core properties, noting this is standard practice described as "a backbone" of recent progress \[Lee et al. 2024\] (36hS-W\_S.6, Appendix D5)
> * **Figure improvements**: Enhanced readability of all figures, particularly original Figure 5 (QFCq-W5)
>
> ## **Closing**
>
> We have comprehensively addressed every concern raised by the reviewers through substantial revisions, additional experiments, and improved presentation. Our work provides essential infrastructure—unified representation, large-scale dataset, and flexible benchmark—for an important unsolved problem at the intersection of ML and materials science. The revised manuscript demonstrates both the feasibility of VLM-based 3D metamaterial design and actionable insights for the broader ML community. We believe these revisions have significantly strengthened the paper and directly address all reviewer concerns.
>
> Again, we sincerely thank all reviewers and ACs -- we appreciate your time, insights, and careful consideration of our manuscript!

---

### Meta-Review · Area_Chair_XWNF · 2026-01-05

**Summary:**

The authors have made a significant effort during rebuttal. I believe most of the concerns are mentioned in the rebuttal.
However, the writing, the exposition and illustrations are not fully accessible for researchers in ML community. Readers would expect the main paper to center on ML-specific elements (such as what is the problem formulation, what is the data structure, how is it different from other 3D data, what is the challenge, why would existing techniques fail). Although the draft has improved during the rebuttal, the paper requires significant restructuring to become ML-friendly. Thus I would recommend rejecting the paper (I would encourage the authors to revise the paper with presentation issues for future submission).

**Reviewer Concerns:**

Reviewer qTbH: is mainly skeptic about the motivation (why MetaDSL? why voxels?). The authors showed some advantages over voxel-based representations.

Reviewer QFCq: missing baselines and weak comparisons. The authors showed additional metrics.

Reviewer vMGX: paper might not be suitable for ML conferences like ICLR. The authors restructured paper and minor revised the paper.

Reviewer 36hS: similar to QFCq, weak comparisons.

**Reviewer Scores:**

The authors did a great job to answer all the questions. Based on my experience, most reviewers would increase the scores a little bit, which makes the scores are borderline.

---

### Decision · Program_Chairs · 2026-01-26

Reject